EMBO
reports

# The spectrum of lysosomal stress and damage responses: from mechanosensing to inflammation

Ori Scott[1,2,3], Ekambir Saran [1] & Spencer A Freeman [1,4]✉

## Abstract

Cells and tissues turn over their aged and damaged components in order to adapt to a changing environment and maintain homeostasis. These functions rely on lysosomes, dynamic and heterogeneous organelles that play essential roles in nutrient redistribution, metabolism, signaling, gene regulation, plasma membrane repair, and immunity. Because of metabolic fluctuations and pathogenic threats, lysosomes must adapt in the short and long term to maintain functionality. In response to such challenges, lysosomes deploy a variety of mechanisms that prevent the breaching of their membrane and escape of their contents, including pathogen-associated molecules and hydrolases. While transient permeabilization of the lysosomal membrane can have acute beneficial effects, supporting inflammation and antigen cross-presentation, sustained or repeated lysosomal perforations have adverse metabolic and transcriptional consequences and can lead to cell death. This review outlines factors contributing to lysosomal stress and damage perception, as well as remedial processes aimed at addressing lysosomal disruptions. We conclude that lysosomal stress plays widespread roles in human physiology and pathology, the understanding and manipulation of which can open the door to novel therapeutic strategies.

**Keywords** Phagosolysosome; Glycocalyx; Host–pathogen; Pore-forming Toxins; Autophagy
**Subject Categories** Membranes & Trafficking; Microbiology, Virology & Host Pathogen Interaction; Organelles

## Introduction

Lysosomes are found in all nucleated mammalian cells and are well known for their capacity to degrade macromolecules imported by endocytosis and intracellular components targeted by autophagy. To do so, these membrane-bound organelles possess a unique composition: they are endowed with more than 60 luminal hydrolases that include proteases, lipases, nucleases, and glycosidases which are made and kept active at an acidic pH (~4.5–5) (Xu and Ren, 2015). The lysosomal $H^+$ gradient is stringently maintained by the action of the V-ATPase that, in addition to providing the acidic environment of the lysosomal lumen, generates the $H^+$-motive force that propels secondary transport pathways (Freeman et al, 2023). Lysosomes also harbor lipid-binding proteins that serve as co-factors for luminal hydrolases by extracting lipids and solubilizing membranous contents (Fürst and Sandhoff, 1992; Matsuda et al, 2004; Morimoto et al, 1989). To protect their own membrane from degradation, lysosomes express a unique set of resident integral glycoproteins—a shielding lysosomal glycocalyx that creates electrostatic and steric barriers that curtail access to the lumen-facing lipids they shroud (Carlsson et al, 1988; Kundra and Kornfeld, 1999; Neiss, 1984; Wilke et al, 2012). While their secluded catabolic reactions are a defining biological role of lysosomes, these organelles are increasingly appreciated as being remarkably dynamic and versatile, contacting and/or merging with other organelles to support a wide variety of physiological processes including in membrane repair, metabolism, cell signaling, pathogen killing, and antigen presentation. To execute such diverse and complex functions, lysosomes are heterogeneous in their size, shape, and position in cells, and are constantly remodeled to undergo fission and fusion with other organelles when needed. The resultant hybrid organelles, including autolysosomes, phagolysosomes, and macropinolysosomes, are also subject to dynamic remodeling. In all cases, lysosomes and lysosome-related organelles must contend with threats to the integrity of their membrane in order to remain acidic and retain their contents.

### Physiologic fluctuations in lysosomal stability

The threats to lysosomal integrity are numerous and occur in both physiological and pathophysiological settings. Everyday hazards include abrupt shifts in the osmolarity of their luminal fluid, changes to their volume and membrane tension, mechanical pressure, and friction forces. Metabolic challenges to the lysosome are in fact frequent: while the ionic composition of the lysosome, notably pH, is stringently regulated, the concentration of organic solutes is in constant flux. As cargo and fluid-phase contents internalized by endocytosis or sealed off from the cytosol by autophagosomes are degraded, lysosomes must contend with fluctuations in osmolarity and resultant changes in volume and endomembrane tension (Saric and Freeman, 2020, 2021). The liberation of sugars, nucleosides, amino acids, and phosphates all

[1]Program in Cell Biology, Peter Gilgan Centre for Research and Learning, Hospital for Sick Children, Toronto, ON, Canada. [2]Division of Clinical Immunology and Allergy, Hospital for Sick Children, Toronto, ON, Canada. [3]Department of Paediatrics, University of Toronto, Toronto, ON, Canada. [4]Department of Biochemistry, University of Toronto, Toronto, ON, Canada. ✉E-mail: spencer.freeman@sickkids.ca

stand to contribute to acute increases in osmolarity that oblige the influx of water. In turn, this may cause increases in membrane tension if not offset by active processes such as the addition of surface area. Additional surface membrane can be acquired through fusion with other organelles, be generated by the acquisition of lipids delivered from the ER via transporters, or be released by curvature-generating/stabilizing protein coats that hold membrane reserves. The degradation of organic polymers also stands to impact the prevailing pH; many of the solutes liberated during this process contain functional groups that can sponge up $H^+$ (Freeman et al, 2023). Once hydrolyzed into transportable units, these solutes are therefore extruded into the cytosol by resident transporters, with an accompanying change in the buffering power of the lysosome and the exit of water. Not surprisingly, defects in lysosomal transport pathways cause osmotic imbalances and can alter the buffering power of the lysosome (Platt et al, 2018). Under conditions where the buffering power increases, the V-ATPase needs to work harder/ longer to achieve the low pH required for optimal lysosomal function (Durgan and Florey, 2022; Sava et al, 2024). Longer-term accommodations to the lysosomal membrane that alleviate high tension like the de novo biosynthesis of lysosome-associated genes may also be required (Sardiello et al, 2009).

## Unique challenges to the lysosomes of myeloid cells

Additional threats to lysosomal integrity emerge in pathological settings including the accumulation of poorly digestible particulates, the intracellular growth of pathogens, and the attack of the lysosomal membrane by pore-forming effectors or microbial enzymes. These insults are most readily encountered by cells of the innate immune system including macrophages and dendritic cells that perform continuous surveillance of their surroundings, where they interface with pathogens. Some of the antimicrobial defenses deployed by these cells, such as the production of reactive oxygen species (ROS), can themselves challenge lysosomal integrity. In addition to piecemeal forms of endocytosis that involve clathrin and caveolae, myeloid cells are endowed with highly embellished and dynamic membranes replete with specialized receptors that propel the internalization of extracellular fluid in bulk via macropinocytosis (Swanson and Watts, 1995) and larger macromolecules (>0.5 μm) by phagocytosis (Gordon, 2016; Rosales and Uribe-Querol, 2017). Macropinocytosis and phagocytosis by these specialized immune cells play diverse and far-reaching roles in tissue homeostasis and monitoring, including the processing of nutrients, cellular by-products and debris, as well as clearance of damaged or dead cells ("efferocytosis") (Doran et al, 2020). Excessive uptake of nutrients via macropinocytosis and phagocytosis, however, can have detrimental consequences. When particulates taken up are not completely digested, the deposition of substrates in their lysosomes places these cells as central instigators of inflammation and pathology (Becker et al, 2010; Dostert et al, 2008; Duewell et al, 2010; Martin et al, 2009; Rajamäki et al, 2010). In atherosclerosis, for example, the uptake of native or oxidized low-density lipoproteins, whether by endocytosis or micropinocytosis, causes lipid (mainly cholesterol) build-up in the macrophage lysosome, resulting in their conversion to foam cells in the vascular intima (Duewell et al, 2010; Moore and Tabas, 2011). This leads to the recruitment of monocytes, further processing of abundant arterial plaques by phagocytosis, and more foam cell formation, a

feed-forward effect. In gout, monosodium urate (MSU) crystals are deposited in joints and internalized by synovial macrophages, where they are accrued by lysosomes and render the cells highly inflammatory (Martin et al, 2009; So and Martinon, 2017). Several neurodegenerative conditions, including Alzheimer's, Parkinson's, and Huntington's disease, all lead to the abnormal synthesis of protein fibrils which accumulate in the lysosomes of microglia, specialized resident phagocytes in the brain, and trigger neuroinflammation (Stancu et al, 2019; van Olst et al, 2020; Wu et al, 2021). Clearly the relationship between lysosomal stress and cell-driven inflammation by overladen myeloid cells in these diseases holds both academic interest and therapeutic potential, which we discuss in detail below.

## Pathogens and their effectors

Pathogens present additional threats to lysosomes in phagocytes. While most microorganisms are effectively killed by the hostile acidic, oxidative and hydrolytic environment of the macropino- and phagolysosomes in these cells, a remarkable number of pathogens have developed means to survive and often grow inside professional phagocytes (Flannagan et al, 2012; Sarantis and Grinstein, 2012). Evasion strategies include compromising the activity and integrity of acidic organelles to either subvert or escape the compartment. In extreme cases, pore-forming toxins, typically utilized by Gram-positive bacteria, rapidly alter the osmotic balance of phagosomes leading to their rupture. *Listeria monocytogenes* releases Listeriolysin O (LLO), a cholesterol-dependent cytolysin with a low optimal pH (~6). LLO cooperates with phospholipases to breakdown endomembranes (Beauregard et al, 1997; Birmingham et al, 2008; Schnupf and Portnoy, 2007). This enables bacterial replication in macrophage vacuoles, followed by escape into the cytosol without inflicting damage to the plasma membrane (Meyer-Morse et al, 2010; Birmingham et al, 2008; Dramsi and Cossart, 2002). Other examples include α-hemolysin from *Staphylococcus aureus* (López de Armentia et al, 2017) and Pneumolysin (PLO) secreted by *Streptococcus pneumoniae* (Inomata et al, 2020), both of which damage the lysosomal membrane to induce organelle swelling.

Gram-negative bacteria such as *Salmonella enterica* or *Shigella flexneri* use a Type 3 Secretion System to invade host cell membranes and subsequently damage and escape their established bacterial vacuoles (Ellison et al, 2020). Vacuolating toxin A (VacA) is a major virulence factor produced by *Helicobacter pylori* which colonizes the gastric mucosa (Terebiznik et al, 2006). VacA forms endomembrane anion channels, causing influx of $Cl^-$, $HCO_3^-$ and small organic anions (Foegeding et al, 2016), leading in turn to osmotic swelling and formation of multivesicular vacuoles where bacteria replicate (Genisset et al, 2007; Amieva et al, 2002). Another recently described toxin secreted by the Gram-negative bacteria *Vibrio cholerae*, called MakA, is also thought to generate pores in acidic organelles with similar effects on membrane traffic (Jia et al, 2022b). In every case, endomembrane perforations induced by bacterial effectors compromise lysosome function and stimulate repair pathways described later.

In addition to destabilizing endomembranes with toxins and effectors, some pathogens can grow within the phagosome to cause its mechanical rupture. For example, *C. albicans* can grow inside phagosomes at rates of 10 μm/h, and while lysosomes continue to

fuse with the growing phagosome, rupture of the limiting membrane inevitably ensues (Westman et al, 2018). Other pathogens achieve the very opposite effect by subverting the endocytic pathway of tissue-resident phagocytes, preventing fusion of the phagosome with lysosomes, and quelling stress-induced inflammation to cloak themselves from immune attack. Mycobacteria such as *M. tuberculosis* demonstrate particular mastery of this skill, as they evade host immune recognition for years while remaining latent inside macrophage endosomes (de Chastellier, 2009). The bacterium achieves this by deploying effectors that target the signaling lipids involved in maturation, thereby preventing the fusion of phagosomes with lysosomes. For these reasons, some highly specialized phagocytes such as conventional type 1 dendritic cells (cDC1) intentionally perforate their endolysosomal membrane to ensure leakage of peptides for antigen cross-presentation and the activation of cytosolic pattern recognition receptors (Gonzales et al, 2024; Rodríguez-Silvestre et al, 2023).

## Experimental systems and approaches that cause lysosome damage

Given the importance of lysosomal stress responses and its sequelae, a number of experimental models have been used to study the process in vitro. One commonly employed reagent, glycyl-L-phenylalanine 2-naphthylamide (GPN), is a membrane-permeant cathepsin substrate; unlike GPN itself, its degradation products cannot diffuse out of the lysosome. This creates a hyperosmolar lumen, and the consequent influx of water can cause rapid loss of lysosome membrane integrity (Berg et al, 1994; Chen et al, 2024; Jadot et al, 1984; Niekamp et al, 2022; Skowyra et al, 2018; Tan and Finkel, 2022). Another popular lysosomotropic agent, leucyl-L-leucine methyl-ester (LLOMe), which also undergoes cathepsin-dependent processing in the lysosome, generates membranolytic polymers, very reproducibly leading to rupture of the organelle (Bonet-Ponce et al, 2020; Bussi et al, 2022; Tan and Finkel, 2022). In phagocytic cell types, other frequently utilized models to induce (phago)lysosomal rupture include crystals such as silica, monosodium urate (MSU) or cholesterol aggregates (Hornung et al, 2008), and pathogens including *Listeria monocytogenes* or just the application of listeriolysin. These models have been instrumental in characterizing the cellular response to overt lysosomal rupture. As lysosomes break open, the release of ions and hydrolases into the cytosol have been proposed to cause extensive organellar damage, NLRP3 inflammasome activation and cell death (Boya and Kroemer, 2008; Kavčič et al, 2017; Hornung et al, 2008; Rajamäki et al, 2010; Joshi et al, 2015; Martinon et al, 2006). While complete lysosomal breakage is highly inflammatory and certainly contributes to pathology, only a few clear examples of lysosome rupture have emerged in vivo (see Box 1).

It stands to reason that lysosomes must have robust mechanisms in place to cope with substantial stress without fully rupturing, even under certain pathological conditions. Whereas complete lysosomal breakage is generally detrimental to cells, lysosomal stress and even transient local ruptures can be deleterious or desirable to the organism, depending on their context, extent and chronicity. With this paradigm in mind, our review explores the spectrum of stress perception and responses in lysosomes (Fig. 1). First, we provide a brief description of mechanisms that protect lysosomes from stress.

---

**Box 1. Lysosomal damage and repair in vivo**

Monitoring lysosome stress, damage, and subsequent repair pathways in vivo presents significant challenges, primarily owed to (i) the difficulty in establishing sufficient resolution of lysosomes in optically tractable tissues and (ii) the lack of reporter systems/models that can be used to monitor these events, dynamically. This is further complicated by adverse photodamaging effects introduced by light microscopy approaches that stand to severely compromise measurements of endogenous membrane damage. Methods for monitoring subtle lysosomal damage/stress include measuring lysosomal pH using ratiometric dyes delivered by fluid phase through the endocytic pathway. Such an approach is challenging in vivo, especially in devising means to build a standard curve for precise pH determinations. Additionally, the dye-conjugated polymers (i.e., dextrans) used to report pH changes can themselves leak out of the lysosome in response to stress, making such measurements exceedingly difficult and negating the opportunity to determine repair mechanisms that ultimately restore acidic pH.

Instead, the visualization of galectin-3 on lysosomes, which accumulates only upon exposure of the glycocalyx, has emerged as a primary method for studying lysosomal rupture in vivo, both in live animals and fixed tissues (Jia et al, 2020c). This approach has uncovered certain genetic and chemical means that compromise lysosomal integrity. For example, genomic deletions in *C. elegans* that disrupt the transport and degradative properties of lysosomes have been shown to lead to galectin-3 targeting to lysosomes in vivo. Specifically, the deletion of CLH-6, the homolog of human ClC-7, which is important for optimal hydrolase activity, as well as SCAV-3, the homolog of human LIMP-2, which is responsible for cholesterol egress from the lysosome, both cause the formation of galectin-3 puncta associated with ruptured lysosomes (Li et al, 2016; Zhang et al, 2023). When delivered to the lungs of mice, silica crystals evoke sufficient lysosomal damage in alveolar macrophages to also cause the recruitment of galectin-3 to the damaged lysosomes (Bussi et al, 2022). The lysosomes of renal epithelial cells suffer a similar fate when challenged with oxalate crystals, an effect also observed in patients suffering from crystal nephropathies (Nakamura et al, 2020). Another means used to measure of lysosomal permeability in vivo is the release of their cathepsins into the cytosol. Hydroxyapatite crystals have been reported to cause lysosomal membrane permeabilization as judged by the loss of the appearance of punctate cathepsin staining in chondrocytes in a murine osteoarthritis model (Ye et al, 2023).

While these studies highlight mechanisms of lysosomal damage/repair and possibilities for its investigation in vivo, most of the research in the field to date has been conducted in vitro. This underscores the need for further investigation with a focus on the dynamics of the endocytic pathway in more complex in vivo settings. Three-dimensional reconstructions of these organelles to catch (transient) perforations, for example, using volumetric focused ion beam scanning electron microscopy, is another attractive possibility (Sanyal et al, 2025) but requires considerable time and effort and needs to be combined with live imaging. Conducting such studies will undoubtedly provide a deeper and more complete understanding of lysosome damage and repair pathways in conditions that more closely resemble physiological states, ultimately advancing our knowledge of lysosomal behavior and its implications in health and disease.

---

Second, and in more detail, we describe the acute responses to lysosomal stress that prevent damage to their membrane including mechanosensing channels, ion conductance, volume regulation, and the addition of surface membrane by fusion and lipid delivery from the ER. Third, we describe the immediate responses triggered by lysosomal damage including the recruitment of scaffolds, stress granules, and autophagic machinery that (attempt to) contain the injury and restore homeostasis. This is followed by an overview of the responses to sustained lysosomal stress that can occur under

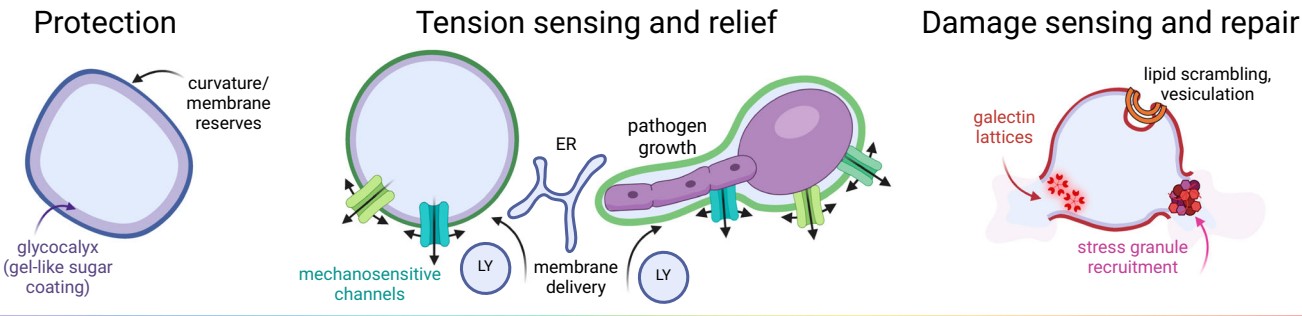

**Figure 1.  The spectrum of lysosome resilience and damage.**

The lysosomal membrane is protected from damage by (i) protein coats that deform its surface to generate curvature as well as (ii) a protective glycocalyx layer that buffers exposure of the membrane from its luminal contents. Upon increases to membrane tension, the lysosome has resident mechanosensitive channels that conduct ions outward with pleiotropic, protective effects including the recruitment of more membrane. Should damage occur, repair mechanisms are elicited, including the formation of galectin lattices, lipid scrambling and vesiculation, and stress granule formation.

pathological conditions, including transcriptional changes, and tissue remodeling. Finally, beneficial aspects of spatiotemporally-controlled lysosomal stress and the mechanisms involved are considered. Given the special role of lysosome stress in myeloid cells in a myriad of physiologic and pathophysiologic conditions, we focus some of our attention on lysosome resilience and responses in these cell types; however, the mechanisms described here have been found in various other cell types where they equally apply.

## Mechanisms that protect lysosomes from stress

Lysosomes are intrinsically protected from membrane damage by their associated proteins that either (i) form a network of luminal-facing polysaccharides with their glycan modifications to produce a "sugar coating" or glycocalyx or (ii) secure readily deployable membrane reserves (Fig. 2). Lysosomal glycoproteins, notably those of the LAMP family, are sufficiently ubiquitous and highly expressed that they are frequently used as standard markers of late organelles in the endocytic pathway (Saftig and Klumperman, 2009). LAMP-1 and LAMP-2 represent 0.1–0.2% of the total cell protein in normal cells and a much larger percentage in highly metabolic or phagocytic cell types (Chen et al, 1985). LAMP proteins are essential for life and required for basic lysosome functions, including their fusion with phagosomes and autophago-somes; these organelles accumulate in their absence, causing neurological defects and muscle weakness (Tanaka et al, 2000). Incredibly, the weight of the glycans in LAMP-1/-2 exceeds that of the core polypeptide chain. They contain 16–18 N-linked glycans, including high-molecular-weight poly-N-acetyl-lactosaminoglycans normally found in mucins, as well as O-linked glycans (Wilke et al, 2012). Other members of the LAMP family are more restricted in their tissue expression and are particularly enriched in myeloid cells, becoming further

upregulated upon infection (Rabinowitz and Gordon, 1991). These include dendritic cell (DC)-LAMP and macrosialin, which are endowed with dozens of O-glycans in their most N-terminal (luminal-facing) regions, making them even more highly glycosy-lated than LAMP-1 and -2 (Fig. 2).

The gel-like glycan layer would in principle protect the underlying membrane from lipases and lipid-extracting proteins, serving as a diffusion barrier and gatekeeper to close encounters with shorter transmembrane proteins and the lipid bilayer. There is remarkably little known, however, about the permeability of the lysosomal glycocalyx and its capacity to restrict molecular diffusion based on size or charge, especially when compared to the established rheological impacts of the plasma membrane glycoca-lyx. While the removal of LAMP-family proteins can cause pathology (Andrejewski et al, 1999; Eskelinen et al, 2004; Tanaka et al, 2000), there must be additional glycoproteins or glycolipids that help to protect lysosomes. This would explain why the limiting membranes of lysosomes in LAMP-1/-2 double-knockout cells appear to be intact. Finally, while the lysosomal glycocalyx likely serves as an important protective barrier, abnormally elevated glycosylation of LAMP-1 also plays a role in pathology, contribut-ing to Niemann-Pick disease for example (Cawley et al, 2020; Kosicek et al, 2018). It follows that removing glycans on LAMP proteins by delivering endoglycosidases through fluid-phase endocytosis or preventing LAMP glycosylation altogether can increase the efficiency of cholesterol export from the lysosome, alleviating its build-up in the lumen in such settings (Chadwick et al, 2024; Li et al, 2015).

Like all mammalian membranes, the membrane bilayer of lysosomes cannot stretch more than even 5%, leaving little room to expand if perfectly spherical. Imaging by transmission electron microscopy reveals that lysosomes, and especially late endosomes, rarely appear as perfect circles in micrographs: the organelles are instead often irregular in shape. This provides an additional protective feature of lysosomes and is facilitated by cytosolic, lipid-binding proteins that deform their limiting membranes. Such

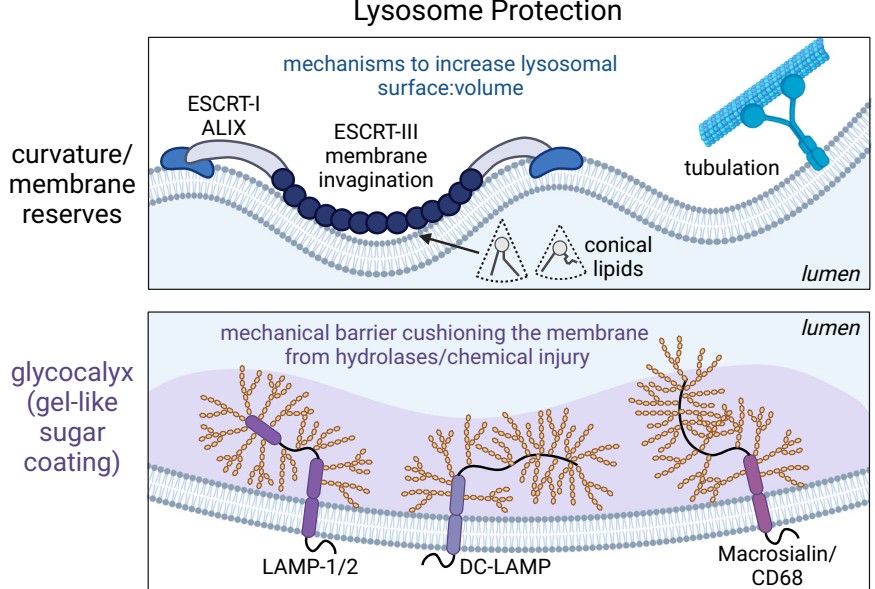

**Figure 2. Lysosome protection.**

Rather than appearing as perfect spheres, lysosomes can assume various shapes and, in some cases, form tubular networks. The connections between lysosomes and the microtubule cytoskeleton, and perhaps protein coats, provide local forces that generate such extreme membrane deformation. Dimpling of the membrane is more common across cell types and is supported by ESCRT proteins as well as lipids that accommodate or generate curvature. Conical lipids like PE likely feature in this effect. Additional protective mechanisms are found on the luminal-facing side of the lysosomal membrane which is protected by its glycocalyx, a gel-like sugar coating. While numerous, the defining members of the lysosomal glycocalyx are the LAMP family of transmembrane proteins that are tailored in their expression and glycosylation according to the microenvironment and feature in disease states.

proteins include the endosomal sorting complexes required for transport (ESCRT) machinery described in detail in other sections. ESCRTs bend the membrane inward, a process that requires coordination with the negatively charged lipid lysobisphosphatidic acid (LBPA)—also known as bis(monoacylglycerol) phosphate (BMP)—found in the luminal leaflet of (endo)lysosomes (Larios et al, 2020). Importantly, recent work demonstrated that the $Ca^{2+}$-dependent recruitment of ESCRT to lysosomes experiencing increased tension prevented their rupture (Chen et al, 2024). While the mechanism(s) by which ESCRT complexes protect the membrane from damage are not yet clear, it is conceivable that the deformation of the membrane under high stress/tension facilitates contacts with other organelles and/or fusion. Conically shaped lipids like phosphatidylethanolamine (PE) and PE-plasmalogens may similarly accommodate high curvature (Fig. 2). In cases when this does not cause complete membrane scission to form intraluminal vesicles, deformation would increase the surface:volume of the compartment, ensuring that membrane reserves are available for rapid volume expansion when necessary. It follows that ESCRTs broadly protect against lysosome damage (Chen et al, 2024).

In a completely different morphological configuration, lysosomes can appear as tubular networks. It remains to be determined what facilitates tubulation in these cases, but the tubes and networks appear to be pulled along microtubules and are accentuated by stimulation of cells with pathogen-associated molecules like LPS (Saric et al, 2016). The initially high surface:volume ratio of tubular lysosomes supplies membrane slack that could be mobilized quickly to prevent gains in membrane

tension and rupture, providing another clear mechanism of protection against damage.

## Acute responses to lysosomal stress that prevent damage

Provided that membrane reserves are not sufficient to accommodate changes to lysosomal volume caused by the insults described above, the limiting membrane of the organelle will begin to incur increased tension, proportional to the application of hydrostatic or pathogen-induced pressure against the membrane. It has very recently become appreciated that such increases in endomembrane tension can be directly perceived by mechanosensitive channels resident in lysosomes, which convert mechanical stimuli into the outward transport of ions. Whereas several plasmalemmal mechanosensitive non-selective cation channels are now well-established (Kefauver et al, 2020), only two (endo)lysosomal counterparts have been characterized: transient receptor potential channel of the mucolipin subfamily 2 (TRPML2) (Chen et al, 2020), and TMEM63A (Li et al, 2024). The presence of a mechanosensitive anion channel in lysosomes is less clear, though components of the Volume-Regulated Anion Channel (VRAC), which is activated on the cell surface by swelling, have been detected in the endocytic pathway (Li et al, 2020).

Like the mechanosensitive cation channels in the plasma membrane, there are likely multiple, convergent inputs that effectively 'gate' open the lysosome-resident cation channels. For example, the gating of all TRPML family members (TRPML1-3)

## Lysosome tension sensing and relief

**Figure 3.   Increased membrane tension and relief.**

Upon high hydrostatic pressure, pressure from growing pathogens, or lipid oxidation, the lysosomal membrane experiences increased tension. Mechanosensitive channels respond, conducting cations outward including $H^+$, $Na^+$, and $Ca^{2+}$. $Ca^{2+}$ is thought to mediate the activation of PI4 kinases (PI4K2A) to generate PI4P, which facilitates membrane contact sites between the stressed lysosomes and the endoplasmic reticulum (ER). Oxysterol-binding protein (OSBP)-related proteins (ORPs) facilitate these contacts and may also exchange lipids between the two organelles, which is more pronounced when ATG2 is recruited. Lysosomal stress also causes thinning and disruption of the membrane bilayer and the engagement of VPS13C, a bridge-like lipid transfer protein. VPS13C creates a tunnel between the stressed lysosome and the ER, independent of ORPs, to also transport lipids. Membrane fusion may also be encouraged by the release cations. Collectively, these pathways increase the surface available for the lysosome to expand and could offset the increases in tension. Alternatively, the outward transport of cations, if paired with anions, could result in volume decrease.

requires phosphatidylinositol 3,5-bisphosphate ($PtdIns(3,5)P_2$), generated via the phosphorylation of phosphatidylinositol 3-phosphate ($PtdIns(3)P$) by the kinase PIKfyve (Dong et al, 2010). The mechanical stimulation of TRPML2 is in fact dependent on the recruitment of $PtdIns(3,5)P_2$ to its binding pocket which is influenced by charge-based interactions, though other domains are likely also involved (Chen et al, 2020). Interestingly, PIKfyve activity and hence the levels of $PtdIns(3,5)P_2$ can be affected by an array of environmental factors including osmotic stress, reactive oxygen species (ROS), hormones, and growth factors (Rivero-Ríos and Weisman, 2022; Bonangelino et al, 2002). The expression of TRPML2 is also significantly upregulated by microbial-derived molecules. Therefore, the levels of $PtdIns(3,5)P_2$ and TRPML2, as tailored by the cellular environment, stand to adjust the mechanosensitivity of lysosomes. At present, little is known about the context-dependent regulation of TMEM63A.

Once mechanosensitive channels are gated by increases in membrane tension, their outward transport of ions from the lysosome lumen to the cytosol has at least three effects that could provide resilience to lysosomal damage, each in response to individual cation currents (Fig. 3). First, they may play a role in the regulatory volume decrease (RVD) of lysosomes. Following an increase in hydrostatic pressure, the outward transport of alkali cations could in principle drive the exit of water and the ensuing shrinkage of lysosomes. To maintain electroneutrality, however, any such RVD response would likely require the efflux of anions with higher concentrations in lysosomes compared to that of the cytosol (i.e., $Cl^-$). In fact, it is likely that the driving force for the directional outward transport of cations and water is the anion current. Enticingly, the cardinal component of VRAC, LRRC8A, has been found to localize also to lysosomes (Li et al, 2020). Supporting the idea that VRAC may be functional in organelles of the endocytic pathway, a recent knockout screen identified

LRRC8A as being necessary to control vacuolar volume; its inactivation resulted in endolysosomal swelling (Li et al, 2020). Outward $Cl^-$ currents have also been detected in lysosomes in response to hypotonic swelling, and the loss of LRRC8A can lead to lysosomal rupture and cell death (Li et al, 2020). Whether or not other anion channels play a role in lysosomal RVD is unclear, and additional investigation of the volume regulation of lysosomes is clearly warranted. It is notable, for example, that at least one member of the Na–K–Cl cotransporter (NKCC) family, SLC12A9, is functional in lysosomes (Accogli et al, 2024; Levin-Konigsberg et al, 2025). SLC12A9 appears to uniquely cotransport ammonium ($NH_4^+$) and $Cl^-$. The former, which accumulates greatly in the highly acidic lumen of lysosomes, could in theory provide an outward driving force for directional transport of solutes and water (Levin-Konigsberg et al, 2025).

A second protective mechanism that would be afforded by mechanosensitive cation channels is their outward transport of $Ca^{2+}$ (Fig. 3). Lysosomes can contain up to mM concentrations of $Ca^{2+}$, which is orders of magnitude higher than the nM concentrations found in the cytosol (Xu and Ren, 2015). Since transient increases in cytosolic $Ca^{2+}$ are measurable and have wide and well-established impacts on cellular signaling, the release of $Ca^{2+}$ from lysosomes has been studied extensively. $Ca^{2+}$ can support the fusogenic activity of SNAREs, triggering their conformational change to promote the addition to lysosomes of membrane surface acquired from other (endo)lysosomes, effectively alleviating high membrane tension. $Ca^{2+}$ release can additionally promote endomembrane remodeling by triggering the recruitment of ESCRT proteins (Chen et al, 2024; Skowyra et al, 2018). While ESCRTs are best appreciated for providing the force that drives the formation of intraluminal vesicles (ILV), their recruitment may generate usable membrane reserves, protecting the organelle from further insults as described above. $Ca^{2+}$ also serves to recruit PI4 kinases (PIK42A) to

generate excess PI4P, which then facilitates membrane contact sites between the stressed lysosomes and the endoplasmic reticulum (ER) (Tan and Finkel, 2022). Oxysterol-binding protein (OSBP)-related proteins (ORPs) in particular facilitate these contacts and exchange lipids between the two organelles. The role of mechanosensitive cation channels and their release of $Ca^{2+}$ in activating protective effects of the PI4P/ESCRT pathways, while not yet established, should be investigated.

Finally, the outward transport of $H^+$ via non-selective cation channels like TMEM63A or TRPML2 could also have acute effects on the protection of lysosomal integrity (Fig. 3). As with $Ca^{2+}$, the outward transport of $H^+$ would be osmotically inconsequential. However, the resultant transient alkalinization of the lysosomal lumen could result in the recruitment of protein complexes that facilitate lysosome remodeling and dynamics. For example, lysosomal alkalinization, sensed directly by the V-ATPase, leads to conformational changes to the pump followed by the recruitment of GTPases of the Rab and Arf families from the cytosol along with their respective guanine-exchange factors (GEFs) (Hurtado-Lorenzo et al, 2006; Maranda et al, 2001; Matsumoto et al, 2022; Wang et al, 2023a). More recently, pH destabilization was shown to recruit the Parkinson's disease-associated kinase, LRRK2, to lysosomes (Eguchi et al, 2018). LRRK2 can then phosphorylate and activate Rabs (Dou et al, 2024; Steger et al, 2016), and also recruit ESCRT proteins (Herbst et al, 2020), adapter proteins (AP-3) (Kuwahara et al, 2016), and actin nucleators to drive membrane remodeling (Bonet-Ponce et al, 2020; Civiero et al, 2018; Kim et al, 2018; Wang et al, 2023b). If the V-ATPase needs to work harder to reacidify the lysosome, this will lead to the conjugation of ATG8 to single membranes, or CASM (Durgan and Florey, 2022); through this process, ATG8 proteins (LC3 and GABARAPs) are conjugated to PS or PE in the endomembrane to then form tubules that engage with ATG2 to promote lipid transfer from the ER to the lysosome (Cross et al, 2023) (Fig. 3). Finally, the disruption of lysosomal acidity or sudden buffering of the compartment also recruits VPS13C (preprint: Wang et al, 2024b), which extends a rod-like tunnel with hydrophobic grooves connecting the ER with (endo) lysosome bilayers (Cai et al, 2022). The C-terminus of VPS13C interacts with Rab7, independent of its phosphorylation, and also interacts specifically with disrupted membranes. Once recruited, the directional transport of lipids to the lysosome via VPS13C would then serve to alleviate tension in the membrane, allowing the limiting membrane to expand, thus protecting the lysosome from undergoing rupture (Fig. 3).

## Cellular responses triggered by lysosomal damage

Should lysosomes exceed their capacity to relieve high membrane tension, rupture to the bilayer may occur, especially if weakened by lipid oxidation (Fig. 4). Overt damage and perforation of lysosomes can be equally inflicted by pore-forming toxins, sharp crystals, or lysosomotropic drugs. Not surprisingly, under various conditions, the damage to lysosomes can range from slight tears, amenable to repair, to complete rupture and bulk loss of luminal contents including hydrolases. Cell-wide breakage of lysosomes is calamitous as active lysosomal hydrolases in the cytosol, notably cathepsins, can instigate cell death (Boya and Kroemer, 2008; Repnik et al,

2012). Accordingly, cells have mechanisms to detect and repair subtle tears to lysosomes, excise membrane pores that are inserted, and identify and eliminate irreversibly damaged lysosomes.

Breaks in the membrane bilayer of lysosomes will cause the instantaneous collapse of their lumen-to-cytosol ion gradients and a sudden drop in membrane tension. As described above, these events effectively recruit and activate ESCRT-III proteins (Skowyra et al, 2018; Mercier et al, 2020; Radulovic et al, 2018). In the plasma membrane, ECSRT has been shown to selectively remove damaging agents such as pore-forming toxins, along with wounded membrane (Jimenez et al, 2014; Wolfmeier et al, 2016). However, this phenomenon remains to be demonstrated for endolysosomes specifically. The high curvature associated with this inward membrane budding requires cytosolically localized neutral sphingomyelinase (nSMase). Following lysosomal damage, $Ca^{2+}$-dependent phospholipid scrambling causes translocation of sphingomyelin (SM) to the cytosolic-facing leaflet, exposing its headgroup to these enzymes (Ellison et al, 2020; Niekamp et al, 2022) (Fig. 4). The phosphorylcholine headgroup of SM is quite large, opposing the generation of high (concave) curvature; scrambled SM can offset the activity of inward budding (Alonso and Goñi, 2018; Niekamp et al, 2022). Upon its hydrolysis by nSMase, however, the scrambled SM is converted to cone-shaped ceramides with small headgroups that accommodate and even drive inward endomembrane budding (Ellison et al, 2020; Niekamp et al, 2022). Such vesiculation can be ESCRT-dependent or possibly also -independent provided these lipids create microdomains with enough negative curvature to bud inward spontaneously (Niekamp et al, 2022). Larger ruptures to the limiting membrane of lysosomes expose their luminal glycocalyx layer and its glycans to cytosolic galectins. In the plasma membrane, the crosslinking of glycoproteins/glycolipids by the pentameric 150 kDa galectin, galectin-3, forms a lattice which impacts molecular diffusion and regulates cell-cell interactions (Boscher et al, 2012; Chiu et al, 2020; Yang et al, 2017). Extrapolating from its plasmalemmal role, it is possible that bridging of lysosomal LAMP-family proteins by galectin-3 may form a barrier that prevents hydrolases from leaving ruptured lysosomes.

Damage to lysosomes also activates the reparative, $Ca^{2+}$-mediated pathways described above that drive the directional transport of lipid from the ER (Tan and Finkel, 2022). By recruiting and activating PI4 kinases, damaged lysosomes are reported to increase PI4P synthesis, which leads to the recruitment of ORPs 9–11 via their pleckstrin homology (PH) domains (Tan and Finkel, 2022). ORPs simultaneously bind to PI4P on the lysosome and to VAPA/B on the ER, effectively causing the ER to wrap around damaged lysosomes. ORPs directly transfer phosphatidylserine from the ER to the lysosome. Conceivably, the close contacts facilitated by ORPs would also facilitate lipid transport by other mediators like VPS13C and ATG2 (Fig. 4).

An even more dramatic temporizing measure for damaged lysosomes is the formation/recruitment of stress granules (Bussi et al, 2023) (Fig. 4). These membraneless organelles can be seen to form de novo in the vicinity of ruptured lysosomes (Jia et al, 2022a), possibly in response to a sudden drop in cytosolic pH or to leakage of lysosomal $Ca^{2+}$ (Bussi et al, 2023; Duran et al, 2024). It is suggested that stress granules then form a stabilizing endomembrane plug, buying time for the recruitment of additional repair machinery such as ESCRT (Bussi et al, 2023). While an attractive

## Lysosome damage sensing and repair

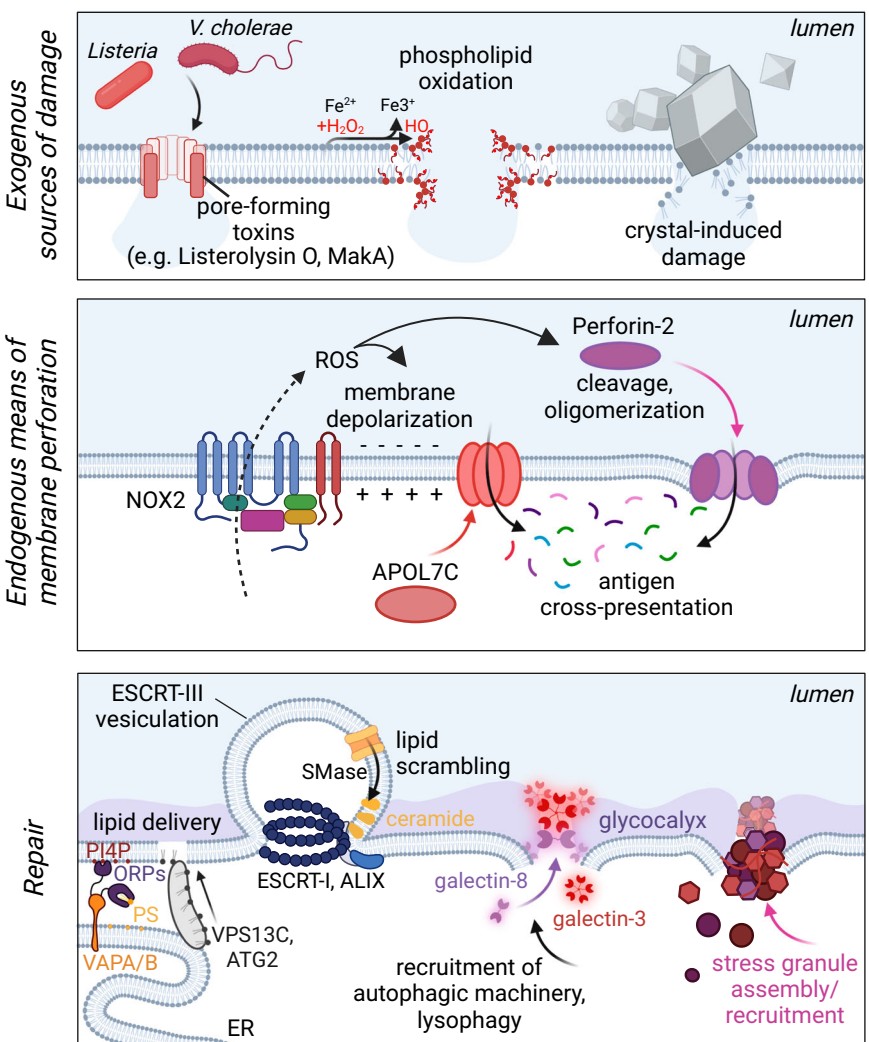

**Figure 4. Lysosomal damage and responses.**

The lysosomal membrane is compromised by pore-forming toxins (as with listeriolysin O or MakA which are secreted by *Listeria* or *V. cholerae*, respectively), oxidation-induced rupturing, and direct destabilization of the membrane caused by pathogens or crystals. Remarkably, specialized cell types like cDC1s can perforate their own endomembranes with endogenously expressed pore-forming proteins like APOL7C and perforin-2. This facilitates the release of peptides and, ultimately, the cross-presentation of antigens. APOL7C tends to insert into depolarized endomembranes, an effect that is potentiated by membrane oxidation via the NADPH oxidase 2 (NOX2). To repair the membrane, embedded toxins can be sorted into vesicles and budded inward by ESCRTs. Certain lipids, including ceramides generated by the hydrolysis of sphingomyelin, help to facilitate the high curvature involved. The exposure of the luminal lysosomal glycocalyx by large breaks in the membrane leads to the recruitment of galectins. Galectins, and other recruited proteins, can then serve to initiate membrane repair or lysophagy depending on the scale of the damage. Finally, stress granules can form in the vicinity of damage to then also participate in membrane repair.

mechanism, further work is required to demonstrate how frequently and reproducibly stress granules form and mobilize to mitigate lysosomal damage.

In cases when all repair mechanisms fail or lysosomes are extensively ruptured, compromised lysosomes are selectively ubiquitinated, leading to their controlled engulfment and degradation by lysophagy (Hung et al, 2013; Thurston et al, 2012; Papadopoulos et al, 2020; Fujita et al, 2013). Key to initiating lysophagy are the aforementioned galectins, galectin-3 and -8 (Fig. 4). After binding luminal glycans, galectin-3 recruits TRIM16, an E3 ubiquitin ligase, which activates autophagy (Chauhan et al,

2016). Galectin-8, on the other hand, targets damaged lysosomes for autophagy by recruiting the cargo receptor NDP52 (Thurston et al, 2012). Galectin-8 is also recruited to sites of exposed cytosolic SM, suggesting a broad spectrum of damage signals recognized by galectins (Ellison et al, 2020). While galectins-3 and -8 have been studied most extensively, other galectins also participate in response to lysosomal damage, including in the initiation of autophagy (Jia et al, 2020a).

Of note, the degree of luminal exposure required to activate lysophagy is substantial, as gaps of even up to 200 nm appear to be amenable to repair (Ellison et al, 2020). This allows for differential

activation of early repair pathways following smaller rupture. The need for ubiquitination as a regulatory gatekeeper, its recognition by receptors, and the ensuing formation of the phagophore makes lysophagy relatively slow. This allows for the more rapid repair pathways to tip the scale in favor of lysosome salvaging in cases of modest damage (Maejima et al, 2013; Fujita et al, 2013).

## Cellular and tissue-wide responses to sustained or repeated lysosomal stress and damage

Alterations to lysosomal osmolarity, hydrostatic pressure or membrane integrity are typically transient, and are rectified within minutes by the mechanisms described above. Should substrates accumulate in lysosomes chronically, either because they are left undigested or are improperly transported, this activates cellular stress responses marked by altered metabolic, signaling and transcriptional pathways. These in turn lead to chronic, low-grade yet cumulative inflammatory effects, which characterize various lysosomal storage diseases (LSD) and neurodegenerative conditions (Aflaki et al, 2016; Cai et al, 2024; Wang et al, 2024a). In contrast, repeated endomembrane damage serves as a potent trigger that results in the assembly of inflammasomes in the cytosol, causing bouts of recurrent inflammation, exemplified by crystal-mediated hyper-inflammatory syndromes (Hornung et al, 2008; Martin et al, 2009; Martinon et al, 2006).

The accumulation of solutes/particles/lipids in lysosomes drastically impairs their ability to transport cargo and recycle nutrients. This loss of functionality is addressed by the de novo biogenesis of lysosomes, orchestrated by activation of the TFEB family of transcription factors (Settembre et al, 2012). Under homeostatic, nutrient-rich conditions, TFEB is inhibited via its phosphorylation by mammalian target of rapamycin complex 1 (mTORC1) (Saxton and Sabatini, 2017; Perera and Zoncu, 2016; Sancak et al, 2010; Bar-Peled and Sabatini, 2014; Settembre et al, 2013). The mTORC1 complex localizes to the surface of lysosomes, where it integrates signals regarding nutrient status (e.g., amino acid availability) via Rag GTPases and their GEFs (Saxton and Sabatini, 2017; Efeyan et al, 2013; Bar-Peled and Sabatini, 2014). During ongoing lysosomal stress, sustained elevated tension on the lysosome membrane can cause the collapse of the lysosomal $Na^+$ gradient via mechanosensitive channels. This ultimately results in an inability to drive the outward transport of cytosolic amino acids in cells where the $Na^+$-coupled amino acid transport from lysosome-to-cytosol is pronounced, including macrophages (Cai et al, 2024). This in turn causes inactivation of mTORC1 followed by dephosphorylation and nuclear translocation of TFEB (Martina et al, 2012; Sardiello et al, 2009). High tension and the accumulation of solutes in lysosomes can also activate TFEB independently of a change in mTORC1 activity (Rusmini et al, 2019). Here, a $Ca^{2+}$ leak from lysosomes may play a role in activating phosphatases that dephosphorylate TFEB (Medina et al, 2015; Rusmini et al, 2019; Sarkar et al, 2007).

Beyond lysosomal biogenesis, the TFEB family of transcription factors are also responsible for cytokine production (Pastore et al, 2016) and can promote tissue recruitment of myeloid cells by inducing the transcription of chemoattractants for monocytes (Cai et al, 2024). In the short-term, compensatory monocyte/ macrophage recruitment may be beneficial to a tissue experiencing lysosomal insufficiency, as these newly recruited cells can assist with alleviating catabolic stress. However, it is anticipated that persistent TFEB-mediated cytokine production would lead to chronic tissue infiltration by more inflammatory macrophages, likely underlying aspects of the progressive pathology seen in LSDs (Cai et al, 2024). Moreover, sustained lysosomal storage renders the endocytic pathway susceptible to leaching out molecules that activate cytosolic pattern recognition receptors. This includes mitochondrial dsDNA acquired via mitophagy which activates the STING pathway (Wang et al, 2024a). The precise mechanisms that permeabilize the lysosomes under these conditions remain to be fully elucidated.

In contrast to high and sustained endomembrane tension, which causes subtle chronic inflammation, the repeated occurrence of lysosomal rupture is overtly inflammatory. The endocytosis of sharp, non-degradable crystals by macrophages leads to recurrent lysosomal damage and spilling of lysosomal cathepsins, the latter potently activating the NLRP3 inflammasome, resulting in pyroptosis (Orlowski et al, 2015; Rajamäki et al, 2010; Hornung et al, 2008; Maejima et al, 2013; Martinon et al, 2006). Even in the absence of inflammasome formation, lysosomal rupture can induce a necrotic type of cell death caused by hydrolases, ROS or protons (Boya and Kroemer, 2008; Kavčič et al, 2017; Gómez-Sintes et al, 2016; Joshi et al, 2015). This is a curious finding in the context of the intermittent nature of inflammatory flares that occur in some crystal-mediated diseases, such as gout. While previous in vitro work suggested regular lysosomal rupture by crystals, in vivo studies suggest that this phenomenon occurs intermittently (Yagnik et al, 2000; Terkeltaub, 2017). Indeed, it has been suggested that crystal-laden lysosomes can undergo repeated cycles of small, resolvable membrane tears without completely rupturing (Joshi et al, 2015; Skowyra et al, 2018). However, the notion of sub-lytic lysosomal damage in crystal-mediated diseases remains to be further explored.

## Beneficial effects of controlled lysosomal damage

Ample evidence demonstrates the deleterious consequences of lysosomal damage. However, emerging literature also reveals important roles for deliberate, timely, and controlled lysosomal perforation in various immune processes, especially in the context of infection. These include antigen cross-presentation by specialized dendritic cells called cDC1s (Canton et al, 2021; Gonzales et al, 2024), and the activation of cytosolic pattern recognition receptors to elicit appropriate inflammation. Pharmacological interventions such as vaccines and lysosomotropic drugs deliberately disrupt lysosomes to achieve these intended effects (Lindblad, 2004; Danielsson and Eriksson, 2021; Pisonero-Vaquero and Medina, 2017). Therefore, understanding how controlled lysosomal damage can be of benefit is compelling.

During infection, the escape of digested peptides from the phagosome serves to enhance microbial recognition by both the innate and adaptive arms of the immune system. In cDC1 cells, peptides need to be released from the phagosome to facilitate antigen processing. This requires import and loading of such peptides onto MHC-I molecules in the ER and traffic to the plasma

membrane to potentiate the activation of cytotoxic T lymphocytes (Colbert et al, 2020). Phagosomal destabilization is initiated by receptors including CLEC9A/DNGR1 that activate the Syk tyrosine kinase and, critically, the NADPH oxidase (Canton et al, 2021). The latter may serve as a trigger for recruitment of endogenously expressed pore-forming proteins, such as apolipoprotein 7c (APOL7C) (Gonzales et al, 2024) or perforin-2 (Rodríguez-Silvestre et al, 2023), allowing for engulfed peptides to be released into the cytosol. Oxidation of the membrane could also facilitate peptide release (Fig. 4). Beyond activation of adaptive T cell responses, induced phagosomal damage is beneficial in enhancing innate recognition by cytosolic pattern recognition receptors. Canonically, microbial di- and tri-peptides are transported from the phagosome to the cytosol by SLC15 family transporters, allowing their binding to NOD1 and NOD2 receptors (Bonham and Kagan, 2014; Hu et al, 2018). However, phagosomal rupture either by endogenous or virulence factors would dramatically scale up this process, allowing for more robust innate immune activation (Bastos et al, 2020).

Deliberate lysosomal rupture mediated by pore-forming gasdermins provides an additional mechanism for controlling and modulating inflammatory pathways. In macrophages, triggering of pyropotosis by the SUMO E3 ligase MAPL causes transfer of mitochondrial DNA (mtDNA) to lysosomes via VPS35. Subsequently, activated gasdermin D and E can be recruited to lysosomes through their lipidation with PS or cardiolipin. Just as they do at the plasma membrane, gasdermins form pores in endomembranes allowing for mtDNA release into the cytosol, triggering activation of dsDNA sensors such as cGAS/STING and at sufficient concentrations, cell death (preprint: Nguyen et al, 2023). Interestingly, in contrast to macrophages, endomembrane targeting by gasdermins in neutrophils has an immune-modulatory effect, which does not trigger cell death. Following NLRP3 inflammasome activation in neutrophils, preferential targeting of LC3+ autophagosomes by gasdermin D drives noncanonical release of IL-1β, without causing lytic cell death (Karmakar et al, 2020).

Lysosomal disruption is frequently utilized pharmacologically. Alum is a common vaccine adjuvant which stimulates inflammatory dendritic cells (Kool et al, 2008), promotes antigen-induced CD4 T cell differentiation and proliferation (Grun and Maurer, 1989; Mannhalter et al, 1985; Serre et al, 2011), and leads to the generation of predominantly IgG1 antibodies (Lindblad et al, 1997). A number of factors may contribute to these effects, including changes in endosomal pH, formation of ROS, disruption of phagosomal membrane stability, and NLRP3 activation (Hornung et al, 2008; Kool et al, 2008). Beyond lysosomal breakage, an additional possibility is that persistence of particulates in endolysosomes could trigger a chronic stress response, accounting for the immune potentiating effects of alum (Danielsson and Eriksson, 2021).

Paradoxically, compounds that act by disrupting lysosomal pH include several anti-inflammatory agents such as macrolide antibiotics and the anti-malarial agent hydroxychloroquine (HCQ), the latter frequently being prescribed for treatment of systemic lupus erythematosus (SLE). Incredibly, the precise anti-inflammatory mechanism of HCQ remains elusive. It is proposed that by inhibiting autophagy, HCQ reduces cytokine release, limits plasma membrane recycling of co-stimulatory molecules, and decreases toll-like receptor activation (Schrezenmeier and Dörner, 2020). However, HCQ activates TFEB/TFE3 which would promote

autophagy (Collins et al, 2021; Schrezenmeier and Dörner, 2020). In addition, lysosomal dysfunction and impaired autophagy are themselves features of SLE (Monteith et al, 2016; Qi et al, 2019). It is therefore clear that much remains to be learned about the precise anti-inflammatory effects of lysosome-modulating drugs.

## Conclusion

The lysosomal stress response involves a multi-faceted set of complementary pathways, allowing for timely modulation of cellular metabolism, signaling, and gene transcription in response to diverse external and endogenous stimuli. The notion of lysosomal resilience in particular provides a finessed paradigm of lysosomal stress, encompassing a frequent and dynamic range of events and outcomes. While the routine, acute insults to lysosomes are swiftly met with homeostatic restoration, sustained changes to endolysosomal integrity trigger widespread chronic effects and induce myeloid-driven inflammation (Box 2). There is a growing appreciation for the contribution of lysosomal rupture to inflammasome activation and cell death, however, its occurrence in vivo is likely intermittent. Instead, baseline inflammation is likely driven by stressed, yet intact, lysosomes. Finally, since spatiotemporally-limited lysosomal stress and rupture are key to various (patho)physiological processes, its pharmacological manipulation stands to have far-reaching benefits for patients.

## Peer review information

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

---

**Box 2. In need of answers**

- Which factors control the activation of mechanosensitive cation channels?

- What provides the driving force for the directional transport of lipids between organelles upon stress/damage? What is the identity of the scramblase exposing SM to the cytosolic endomembrane leaflet following rupture? Is the same scramblase implicated in endomembrane repair, by shuffling phospholipids transferred from the ER?

- To what extent does inflammation play a role in the various systemic manifestations of lysosomal storage disorders?

- Can lysosomal stress or damage be precisely manipulated to generate more efficacious treatments against viral infections, inflammatory diseases or cancer?

Alonso A, Goñi FM (2018) The physical properties of ceramides in membranes. Annu Rev Biophys 47:633–654

Amieva MR, Salama NR, Tompkins LS, Falkow S (2002) Helicobacter pylori enter and survive within multivesicular vacuoles of epithelial cells. Cell Microbiol 4:677–690

Andrejewski N, Punnonen EL, Guhde G, Tanaka Y, Lüllmann-Rauch R, Hartmann D, von Figura K, Saftig P (1999) Normal lysosomal morphology and function in LAMP-1-deficient mice. J Biol Chem 274:12692–12701

Bar-Peled L, Sabatini DM (2014) Regulation of mTORC1 by amino acids. Trends Cell Biol 24:400–406

Bastos PAD, Wheeler R, Boneca IG (2020) Uptake, recognition and responses to peptidoglycan in the mammalian host. FEMS Microbiol Rev 45:fuaa044

Beauregard KE, Lee KD, Collier RJ, Swanson JA (1997) pH-dependent perforation of macrophage phagosomes by listeriolysin O from Listeria monocytogenes. J Exp Med 186:1159–1163

Becker L, Gharib SA, Irwin AD, Wijsman E, Vaisar T, Oram JF, Heinecke JW (2010) A macrophage sterol-responsive network linked to atherogenesis. Cell Metab 11:125–135

Berg TO, Strømhaug E, Løvdal T, Seglen O, Berg T (1994) Use of glycyl-L-phenylalanine 2-naphthylamide, a lysosome-disrupting cathepsin C substrate, to distinguish between lysosomes and prelysosomal endocytic vacuoles. Biochem J 300:229–236

Birmingham CL, Canadien V, Kaniuk NA, Steinberg BE, Higgins DE, Brumell JH (2008) Listeriolysin O allows Listeria monocytogenes replication in macrophage vacuoles. Nature 451:350–354

Bonangelino CJ, Nau JJ, Duex JE, Brinkman M, Wurmser AE, Gary JD, Emr SD, Weisman LS (2002) Osmotic stress-induced increase of phosphatidylinositol 3,5-bisphosphate requires Vac14p, an activator of the lipid kinase Fab1p. J Cell Biol 156:1015–1028

Bonet-Ponce L, Beilina A, Williamson CD, Lindberg E, Kluss JH, Saez-Atienzar S, Landeck N, Kumaran R, Mamais A, Bleck CKE et al (2020) LRRK2 mediates tubulation and vesicle sorting from lysosomes. Sci Adv 6:eabb2454

Bonham KS, Kagan JC (2014) Endosomes as platforms for NOD-like receptor signaling. Cell Host Microbe 15:523–525

Boscher C, Zheng YZ, Lakshminarayan R, Johannes L, Dennis JW, Foster LJ, Nabi IR (2012) Galectin-3 protein regulates mobility of N-cadherin and GM1 ganglioside at cell-cell junctions of mammary carcinoma cells. J Biol Chem 287:32940–32952

Boya P, Kroemer G (2008) Lysosomal membrane permeabilization in cell death. Oncogene 27:6434–6451

Bussi C, Heunis T, Pellegrino E, Bernard EM, Bah N, Dos Santos MS, Santucci P, Aylan B, Rodgers A, Fearns A et al (2022) Lysosomal damage drives mitochondrial proteome remodelling and reprograms macrophage immunometabolism. Nat Commun 13:7338

Bussi C, Mangiarotti A, Vanhille-Campos C, Aylan B, Pellegrino E, Athanasiadi N, Fearns A, Rodgers A, Franzmann TM, Šarić A et al (2023) Stress granules plug and stabilize damaged endolysosomal membranes. Nature 623:1062–1069

Cai R, Scott O, Ye G, Le T, Saran E, Kwon W, Inpanathan S, Sayed BA, Botelho RJ, Saric A et al (2024) Pressure sensing of lysosomes enables control of TFEB responses in macrophages. Nat Cell Biol 26:1247–1260. 1–14

Cai S, Wu Y, Guillén-Samander A, Hancock-Cerutti W, Liu J, De Camilli P (2022) In situ architecture of the lipid transport protein VPS13C at ER–lysosome membrane contacts. Proc Natl Acad Sci USA 119:e2203769119

Canton J, Blees H, Henry CM, Buck MD, Schulz O, Rogers NC, Childs E, Zelenay S, Rhys H, Domart M-C et al (2021) The receptor DNGR-1 signals for phagosomal rupture to promote cross-presentation of dead-cell-associated antigens. Nat Immunol 22:140–153

Carlsson SR, Roth J, Piller F, Fukuda M (1988) Isolation and characterization of human lysosomal membrane glycoproteins, h-lamp-1 and h-lamp-2. Major

sialoglycoproteins carrying polylactosaminoglycan. J Biol Chem 263:18911–18919

Cawley NX, Sojka C, Cougnoux A, Lyons AT, Nicoli E-R, Wassif CA, Porter FD (2020) Abnormal LAMP1 glycosylation may play a role in Niemann-Pick disease, type C pathology. PLoS ONE 15:e0227829

Chadwick SR, Barreda D, Wu JZ, Ye G, Yusuf B, Ren D, Freeman SA (2024) Two-pore channels regulate endomembrane tension to enable remodeling and resolution of phagolysosomes. Proc Natl Acad Sci USA 121:e2309465121

Chauhan S, Kumar S, Jain A, Ponpuak M, Mudd MH, Kimura T, Choi SW, Peters R, Mandell M, Bruun J-A et al (2016) TRIMs and galectins globally cooperate and TRIM16 and Galectin-3 co-direct autophagy in endomembrane damage homeostasis. Dev Cell 39:13–27

Chen C-C, Krogsaeter E, Butz ES, Li Y, Puertollano R, Wahl-Schott C, Biel M, Grimm C (2020) TRPML2 is an osmo/mechanosensitive cation channel in endolysosomal organelles. Sci Adv 6:eabb5064

Chen JW, Pan W, D'Souza MP, August JT (1985) Lysosome-associated membrane proteins: characterization of LAMP-1 of macrophage P388 and mouse embryo 3T3 cultured cells. Arch Biochem Biophys 239:574–586

Chen W, Motsinger MM, Li J, Bohannon KP, Hanson PI (2024) Ca2+-sensor ALG-2 engages ESCRTs to enhance lysosomal membrane resilience to osmotic stress. Proc Natl Acad Sci USA 121:e2318412121

Chiu Y-P, Sun Y-C, Qiu D-C, Lin Y-H, Chen Y-Q, Kuo J-C, Huang J (2020) Liquid-liquid phase separation and extracellular multivalent interactions in the tale of galectin-3. Nat Commun 11:1229

Civiero L, Cogo S, Biosa A, Greggio E (2018) The role of LRRK2 in cytoskeletal dynamics. Biochem Soc Trans 46:1653–1663

Colbert JD, Cruz FM, Rock KL (2020) Cross-presentation of exogenous antigens on MHC I molecules. Curr Opin Immunol 64:1–8

Collins KP, Witta S, Coy JW, Pang Y, Gustafson DL (2021) Lysosomal biogenesis and implications for hydroxychloroquine disposition. J Pharm Exp Ther 376:294–305

Cross J, Durgan J, McEwan DG, Tayler M, Ryan KM, Florey O (2023) Lysosome damage triggers direct ATG8 conjugation and ATG2 engagement via non-canonical autophagy. J Cell Biol 222:e202303078

Danielsson R, Eriksson H (2021) Aluminium adjuvants in vaccines—a way to modulate the immune response. Semin Cell Dev Biol 115:3–9

de Chastellier C (2009) The many niches and strategies used by pathogenic mycobacteria for survival within host macrophages. Immunobiology 214:526–542

Dong X, Shen D, Wang X, Dawson T, Li X, Zhang Q, Cheng X, Zhang Y, Weisman LS, Delling M et al (2010) PI(3,5)P(2) controls membrane trafficking by direct activation of mucolipin Ca(2+) release channels in the endolysosome. Nat Commun 1:38

Doran AC, Yurdagul A, Tabas I (2020) Efferocytosis in health and disease. Nat Rev Immunol 20:254–267

Dostert C, Pétrilli V, Van Bruggen R, Steele C, Mossman BT, Tschopp J (2008) Innate immune activation through Nalp3 inflammasome sensing of asbestos and silica. Science 320:674–677

Dou D, Aiken J, Holzbaur ELF (2024) RAB3 phosphorylation by pathogenic LRRK2 impairs trafficking of synaptic vesicle precursors. J Cell Biol 223:e202307092

Dramsi S, Cossart P (2002) Listeriolysin O: a genuine cytolysin optimized for an intracellular parasite. J Cell Biol 156:943–946

Duewell P, Kono H, Rayner KJ, Sirois CM, Vladimer G, Bauernfeind FG, Abela GS, Franchi L, Nuñez G, Schnurr M et al (2010) NLRP3 inflammasomes are required for atherogenesis and activated by cholesterol crystals. Nature 464:1357–1361

Duran J, Salinas JE, Wheaton RP, Poolsup S, Allers L, Rosas-Lemus M, Chen L, Cheng Q, Pu J, Salemi M et al (2024) Calcium signaling from damaged lysosomes induces cytoprotective stress granules. EMBO J 43:6410–6443

Durgan J, Florey O (2022) Many roads lead to CASM: diverse stimuli of noncanonical autophagy share a unifying molecular mechanism. Sci Adv 8:eabo1274

Efeyan A, Zoncu R, Chang S, Gumper I, Snitkin H, Wolfson RL, Kirak O, Sabatini DD, Sabatini DM (2013) Regulation of mTORC1 by the Rag GTPases is necessary for neonatal autophagy and survival. Nature 493:679–683

Eguchi T, Kuwahara T, Sakurai M, Komori T, Fujimoto T, Ito G, Yoshimura S-I, Harada A, Fukuda M, Koike M et al (2018) LRRK2 and its substrate Rab GTPases are sequentially targeted onto stressed lysosomes and maintain their homeostasis. Proc Natl Acad Sci USA 115:E9115–E9124

Ellison CJ, Kukulski W, Boyle KB, Munro S, Randow F (2020) Transbilayer movement of sphingomyelin precedes catastrophic breakage of enterobacteria-containing vacuoles. Curr Biol 30:2974–2983.e6

Eskelinen EL, Schmidt CK, Neu S, Willenborg M, Fuertes G, Salvador N, Tanaka Y, Lüllmann-Rauch R, Hartmann D, Heeren J et al (2004) Disturbed cholesterol traffic but normal proteolytic function in LAMP-1/LAMP-2 double-deficient fibroblasts. Mol Biol Cell 15:3132–3145

Flannagan RS, Jaumouillé V, Grinstein S (2012) The cell biology of phagocytosis. Annu Rev Pathol 7:61–98

Foegeding NJ, Caston RR, McClain MS, Ohi MD, Cover TL (2016) An overview of *Helicobacter pylori* VacA toxin biology. Toxins 8:173

Freeman SA, Grinstein S, Orlowski J (2023) Determinants, maintenance, and function of organellar pH. Physiol Rev 103:515–606

Fujita N, Morita E, Itoh T, Tanaka A, Nakaoka M, Osada Y, Umemoto T, Saitoh T, Nakatogawa H, Kobayashi S et al (2013) Recruitment of the autophagic machinery to endosomes during infection is mediated by ubiquitin. J Cell Biol 203:115–128

Fürst W, Sandhoff K (1992) Activator proteins and topology of lysosomal sphingolipid catabolism. Biochim Biophys Acta 1126:1–16

Genisset C, Puhar A, Calore F, de Bernard M, Dell'Antone P, Montecucco C (2007) The concerted action of the *Helicobacter pylori* cytotoxin VacA and of the v-ATPase proton pump induces swelling of isolated endosomes. Cell Microbiol 9:1481–1490

Gómez-Sintes R, Ledesma MD, Boya P (2016) Lysosomal cell death mechanisms in aging. Ageing Res Rev 32:150–168

Gonzales GA, Huang S, Wilkinson L, Nguyen JA, Sikdar S, Piot C, Naumenko V, Rajwani J, Wood CM, Dinh I et al (2024) The pore-forming apolipoprotein APOL7C drives phagosomal rupture and antigen cross-presentation by dendritic cells. Sci Immunol 9:eadn2168

Gordon S (2016) Phagocytosis: an immunobiologic process. Immunity 44:463–475

Grun JL, Maurer PH (1989) Different T helper cell subsets elicited in mice utilizing two different adjuvant vehicles: the role of endogenous interleukin 1 in proliferative responses. Cell Immunol 121:134–145

Herbst S, Campbell P, Harvey J, Bernard EM, Papayannopoulos V, Wood NW, Morris HR, Gutierrez MG (2020) LRRK2 activation controls the repair of damaged endomembranes in macrophages. EMBO J 39:e104494

Hornung V, Bauernfeind F, Halle A, Samstad EO, Kono H, Rock KL, Fitzgerald KA, Latz E (2008) Silica crystals and aluminum salts activate the NALP3 inflammasome through phagosomal destabilization. Nat Immunol 9:847–856

Hu Y, Song F, Jiang H, Nuñez G, Smith DE (2018) SLC15A2 and SLC15A4 mediate the transport of bacterially-derived di/tripeptides to enhance the NOD-dependent immune response in mouse bone marrow-derived macrophages. J Immunol 201:652–662

Hung Y-H, Chen LM-W, Yang J-Y, Yang WY (2013) Spatiotemporally controlled induction of autophagy-mediated lysosome turnover. Nat Commun 4:2111

Hurtado-Lorenzo A, Skinner M, El Annan J, Futai M, Sun-Wada G-H, Bourgoin S, Casanova J, Wildeman A, Bechoua S, Ausiello DA et al (2006) V-ATPase

interacts with ARNO and Arf6 in early endosomes and regulates the protein degradative pathway. Nat Cell Biol 8:124–136

Inomata M, Xu S, Chandra P, Meydani SN, Takemura G, Philips JA, Leong JM (2020) Macrophage LC3-associated phagocytosis is an immune defense against Streptococcus pneumoniae that diminishes with host aging. Proc Natl Acad Sci USA 117:33561–33569

Jadot M, Colmant C, Wattiaux-De Coninck S, Wattiaux R (1984) Intralysosomal hydrolysis of glycyl-L-phenylalanine 2-naphthylamide. Biochem J 219:965–970

Jia J, Bissa B, Brecht L, Allers L, Choi SW, Gu Y, Zbinden M, Burge MR, Timmins G, Hallows K et al (2020a) AMPK, a regulator of metabolism and autophagy, is activated by lysosomal damage via a novel galectin-directed ubiquitin signal transduction system. Mol Cell 77:951–969.e9

Jia J, Claude-Taupin A, Gu Y, Choi SW, Peters R, Bissa B, Mudd MH, Allers L, Pallikkuth S, Lidke KA et al (2020c) Galectin-3 coordinates a cellular system for lysosomal repair and removal. Dev Cell 52:69–87.e8

Jia J, Wang F, Bhujabal Z, Peters R, Mudd M, Duque T, Allers L, Javed R, Salemi M, Behrends C et al (2022a) Stress granules and mTOR are regulated by membrane atg8ylation during lysosomal damage. J Cell Biol 221:e202207091

Jia X, Knyazeva A, Zhang Y, Castro-Gonzalez S, Nakamura S, Carlson L-A, Yoshimori T, Corkery DP, Wu Y-W (2022b) V. cholerae MakA is a cholesterol-binding pore-forming toxin that induces non-canonical autophagy. J Cell Biol 221:e202206040

Jimenez AJ, Maiuri P, Lafaurie-Janvore J, Divoux S, Piel M, Perez F (2014) ESCRT machinery is required for plasma membrane repair. Science 343:1247136

Joshi GN, Goetjen AM, Knecht DA (2015) Silica particles cause NADPH oxidase-independent ROS generation and transient phagolysosomal leakage. Mol Biol Cell 26:3150–3164

Karmakar M, Minns M, Greenberg EN, Diaz-Aponte J, Pestonjamasp K, Johnson JL, Rathkey JK, Abbott DW, Wang K, Shao F et al (2020) N-GSDMD trafficking to neutrophil organelles facilitates IL-1β release independently of plasma membrane pores and pyroptosis. Nat Commun 11:2212

Kavčič N, Pegan K, Turk B (2017) Lysosomes in programmed cell death pathways: from initiators to amplifiers. Biol Chem 398:289–301

Kefauver JM, Ward AB, Patapoutian A (2020) Discoveries in structure and physiology of mechanically activated ion channels. Nature 587:567–576

Kim KS, Marcogliese PC, Yang J, Callaghan SM, Resende V, Abdel-Messih E, Marras C, Visanji NP, Huang J, Schlossmacher MG et al (2018) Regulation of myeloid cell phagocytosis by LRRK2 via WAVE2 complex stabilization is altered in Parkinson's disease. Proc Natl Acad Sci USA 115:E5164

Kool M, Pétrilli V, De Smedt T, Rolaz A, Hammad H, van Nimwegen M, Bergen IM, Castillo R, Lambrecht BN, Tschopp J (2008) Cutting edge: alum adjuvant stimulates inflammatory dendritic cells through activation of the NALP3 inflammasome. J Immunol 181:3755–3759

Kosicek M, Gudelj I, Horvatic A, Jovic T, Vuckovic F, Lauc G, Hecimovic S (2018) N-glycome of the lysosomal glycocalyx is altered in Niemann-Pick type C disease (NPC) model cells. Mol Cell Proteom 17:631–642

Kundra R, Kornfeld S (1999) Asparagine-linked oligosaccharides protect Lamp-1 and Lamp-2 from intracellular proteolysis. J Biol Chem 274:31039–31046

Kuwahara T, Inoue K, D'Agati VD, Fujimoto T, Eguchi T, Saha S, Wolozin B, Iwatsubo T, Abeliovich A (2016) LRRK2 and RAB7L1 coordinately regulate axonal morphology and lysosome integrity in diverse cellular contexts. Sci Rep 6:29945

Larios J, Mercier V, Roux A, Gruenberg J (2020) ALIX- and ESCRT-III–dependent sorting of tetraspanins to exosomes. J Cell Biol 219:e201904113

Levin-Konigsberg R, Mitra K, Spees K, Nigam A, Liu K, Januel C, Hivare P, Arana SM, Prolo LM, Kundaje A et al (2025) An SLC12A9-dependent ion transport mechanism maintains lysosomal osmolarity Dev Cell 60:220–235

Li J, Deffieu MS, Lee PL, Saha P, Pfeffer SR (2015) Glycosylation inhibition reduces cholesterol accumulation in NPC1 protein-deficient cells. Proc Natl Acad Sci USA 112:14876–14881

Li K, Guo Y, Wang Y, Zhu R, Chen W, Cheng T, Zhang X, Jia Y, Liu T, Zhang W et al (2024) Drosophila TMEM63 and mouse TMEM63A are lysosomal mechanosensory ion channels. Nat Cell Biol 26:393–403

Li P, Hu M, Wang C, Feng X, Zhao Z, Yang Y, Sahoo N, Gu M, Yang Y, Xiao S et al (2020) LRRC8 family proteins within lysosomes regulate cellular osmoregulation and enhance cell survival to multiple physiological stresses. Proc Natl Acad Sci USA 117:29155–29165

Li Y, Chen B, Zou W, Wang X, Wu Y, Zhao D, Sun Y, Liu Y, Chen L, Miao L et al (2016) The lysosomal membrane protein SCAV-3 maintains lysosome integrity and adult longevity. J Cell Biol 215:167–185

Lindblad EB (2004) Aluminium compounds for use in vaccines. Immunol Cell Biol 82:497–505

Lindblad EB, Elhay MJ, Silva R, Appelberg R, Andersen P (1997) Adjuvant modulation of immune responses to tuberculosis subunit vaccines. Infect Immun 65:623–629

López de Armentia MM, Gauron MC, Colombo MI (2017) *Staphylococcus aureus* alpha-toxin induces the formation of dynamic tubules labeled with LC3 within host cells in a Rab7 and Rab1b-dependent manner. Front Cell Infect Microbiol 7:431

Maejima I, Takahashi A, Omori H, Kimura T, Takabatake Y, Saitoh T, Yamamoto A, Hamasaki M, Noda T, Isaka Y et al (2013) Autophagy sequesters damaged lysosomes to control lysosomal biogenesis and kidney injury. EMBO J 32:2336–2347

Mannhalter JW, Neychev HO, Zlabinger GJ, Ahmad R, Eibl MM (1985) Modulation of the human immune response by the non-toxic and non-pyrogenic adjuvant aluminium hydroxide: effect on antigen uptake and antigen presentation. Clin Exp Immunol 61:143

Maranda B, Brown D, Bourgoin S, Casanova JE, Vinay P, Ausiello DA, Marshansky V (2001) Intra-endosomal pH-sensitive recruitment of the Arf-nucleotide exchange factor ARNO and Arf6 from cytoplasm to proximal tubule endosomes. J Biol Chem 276:18540–18550

Martin WJ, Walton M, Harper J (2009) Resident macrophages initiating and driving inflammation in a monosodium urate monohydrate crystal-induced murine peritoneal model of acute gout. Arthritis Rheum 60:281–289

Martina JA, Chen Y, Gucek M, Puertollano R (2012) MTORC1 functions as a transcriptional regulator of autophagy by preventing nuclear transport of TFEB. Autophagy 8:903–914

Martinon F, Pétrilli V, Mayor A, Tardivel A, Tschopp J (2006) Gout-associated uric acid crystals activate the NALP3 inflammasome. Nature 440:237–241

Matsuda J, Kido M, Tadano-Aritomi K, Ishizuka I, Tominaga K, Toida K, Takeda E, Suzuki K, Kuroda Y (2004) Mutation in saposin D domain of sphingolipid activator protein gene causes urinary system defects and cerebellar Purkinje cell degeneration with accumulation of hydroxy fatty acid-containing ceramide in mouse. Hum Mol Genet 13:2709–2723

Matsumoto N, Sekiya M, Sun-Wada G-H, Wada Y, Nakanishi-Matsui M (2022) The lysosomal V-ATPase a3 subunit is involved in localization of Mon1-Ccz1, the GEF for Rab7, to secretory lysosomes in osteoclasts. Sci Rep 12:8455

Medina DL, Di Paola S, Peluso I, Armani A, De Stefani D, Venditti R, Montefusco S, Scotto-Rosato A, Prezioso C, Forrester A et al (2015) Lysosomal calcium signalling regulates autophagy through calcineurin and TFEB. Nat Cell Biol 17:288–299

Mercier V, Larios J, Molinard G, Goujon A, Matile S, Gruenberg J, Roux A (2020) Endosomal membrane tension regulates Escrt-III-dependent intra-lumenal vesicle formation. Nat Cell Biol 22:947–959

Meyer-Morse N, Robbins JR, Rae CS, Mochegova SN, Swanson MS, Zhao Z, Virgin HW, Portnoy D (2010) Listeriolysin O is necessary and sufficient to induce autophagy during *Listeria monocytogenes* infection. PLoS ONE 5:e8610

Monteith AJ, Kang S, Scott E, Hillman K, Rajfur Z, Jacobson K, Costello MJ, Vilen BJ (2016) Defects in lysosomal maturation facilitate the activation of innate sensors in systemic lupus erythematosus. Proc Natl Acad Sci USA 113:E2142–E2151

Moore KJ, Tabas I (2011) Macrophages in the pathogenesis of atherosclerosis. Cell 145:341–355

Morimoto S, Martin BM, Yamamoto Y, Kretz KA, O'Brien JS, Kishimoto Y (1989) Saposin A: second cerebrosidase activator protein. Proc Natl Acad Sci USA 86:3389–3393

Nakamura S, Shigeyama S, Minami S, Shima T, Akayama S, Matsuda T, Esposito A, Napolitano G, Kuma A, Namba-Hamano T et al (2020) LC3 lipidation is essential for TFEB activation during the lysosomal damage response to kidney injury. Nat Cell Biol 22:1252–1263

Neiss WF (1984) A coat of glycoconjugates on the inner surface of the lysosomal membrane in the rat kidney. Histochemistry 80:603–608

Nguyen M, Collier JJ, Ignatenko O, Morin G, Huang S, Desjardins M, McBride HM (2023) Parkinson's genes orchestrate pyroptosis through selective trafficking of mtDNA to leaky lysosomes. Preprint at https://doi.org/10.1101/2023.09.11.557213 [PREPRINT]

Niekamp P, Scharte F, Sokoya T, Vittadello L, Kim Y, Deng Y, Südhoff E, Hilderink A, Imlau M, Clarke CJ et al (2022) Ca2+-activated sphingomyelin scrambling and turnover mediate ESCRT-independent lysosomal repair. Nat Commun 13:1875

Orlowski GM, Colbert JD, Sharma S, Bogyo M, Robertson SA, Rock KL (2015) Multiple cathepsins promote pro-IL-1β synthesis and NLRP3-mediated IL-1β activation. J Immunol 195:1685–1697

Papadopoulos C, Kravic B, Meyer H (2020) Repair or lysophagy: dealing with damaged lysosomes. J Mol Biol 432:231–239

Pastore N, Brady OA, Diab HI, Martina JA, Sun L, Huynh T, Lim J-A, Zare H, Raben N, Ballabio A et al (2016) TFEB and TFE3 cooperate in the regulation of the innate immune response in activated macrophages. Autophagy 12:1240–1258

Perera RM, Zoncu R (2016) The lysosome as a regulatory hub. Annu Rev Cell Dev Biol 32:223–253

Pisonero-Vaquero S, Medina DL (2017) Lysosomotropic drugs: pharmacological tools to study lysosomal function. Curr Drug Metab 18:1147–1158

Platt FM, d'Azzo A, Davidson BL, Neufeld EF, Tifft CJ (2018) Lysosomal storage diseases. Nat Rev Dis Prim 4:27

Qi Y-Y, Zhou X-J, Zhang H (2019) Autophagy and immunological aberrations in systemic lupus erythematosus. Eur J Immunol 49:523–533

Rabinowitz SS, Gordon S (1991) Macrosialin, a macrophage-restricted membrane sialoprotein differentially glycosylated in response to inflammatory stimuli. J Exp Med 174:827–836

Radulovic M, Schink KO, Wenzel EM, Nähse V, Bongiovanni A, Lafont F, Stenmark H (2018) ESCRT-mediated lysosome repair precedes lysophagy and promotes cell survival. EMBO J 37:e99753

Rajamäki K, Lappalainen J, Öörni K, Välimäki E, Matikainen S, Kovanen PT, Eklund KK (2010) Cholesterol crystals activate the NLRP3 inflammasome in human macrophages: a novel link between cholesterol metabolism and inflammation. PLoS ONE 5:e11765

Repnik U, Stoka V, Turk V, Turk B (2012) Lysosomes and lysosomal cathepsins in cell death. Biochim Biophys Acta 1824:22–33

Rivero-Ríos P, Weisman LS (2022) Roles of PIKfyve in multiple cellular pathways. Curr Opin Cell Biol 76:102086

Rodríguez-Silvestre P, Laub M, Krawczyk PA, Davies AK, Schessner JP, Parveen R, Tuck BJ, McEwan WA, Borner GHH, Kozik P (2023) Perforin-2 is a pore-forming effector of endocytic escape in cross-presenting dendritic cells. Science 380:1258–1265

Rosales C, Uribe-Querol E (2017) Phagocytosis: a fundamental process in immunity. Biomed Res Int 2017:9042851

Rusmini P, Cortese K, Crippa V, Cristofani R, Cicardi ME, Ferrari V, Vezzoli G, Tedesco B, Meroni M, Messi E et al (2019) Trehalose induces autophagy via lysosomal-mediated TFEB activation in models of motoneuron degeneration. Autophagy 15:631–651

Saftig P, Klumperman J (2009) Lysosome biogenesis and lysosomal membrane proteins: trafficking meets function. Nat Rev Mol Cell Biol 10:623–635

Sancak Y, Bar-Peled L, Zoncu R, Markhard AL, Nada S, Sabatini DM (2010) Ragulator-Rag complex targets mTORC1 to the lysosomal surface and is necessary for its activation by amino acids. Cell 141:290–303

Sanyal A, Scanavachi G, Somerville E, Saminathan A, Nair A, Bango Da Cunha Correia RF, Aylan B, Sitarska E, Oikonomou A, Hatzakis NS et al (2025) Neuronal constitutive endolysosomal perforations enable α-synuclein aggregation by internalized PFFs. J Cell Biol 224:e202401136

Sarantis H, Grinstein S (2012) Subversion of phagocytosis for pathogen survival. Cell Host Microbe 12:419–431

Sardiello M, Palmieri M, di Ronza A, Medina DL, Valenza M, Gennarino VA, Di Malta C, Donaudy F, Embrione V, Polishchuk RS et al (2009) A gene network regulating lysosomal biogenesis and function. Science 325:473–477

Saric A, Freeman SA (2020) Endomembrane tension and trafficking. Front Cell Dev Biol 8:611326

Saric A, Freeman SA (2021) Solutes as controllers of endomembrane dynamics. Nat Rev Mol Cell Biol 22:237–238

Saric A, Hipolito VEB, Kay JG, Canton J, Antonescu CN, Botelho RJ (2016) mTOR controls lysosome tubulation and antigen presentation in macrophages and dendritic cells. Mol Biol Cell 27:321–333

Sarkar S, Davies JE, Huang Z, Tunnacliffe A, Rubinsztein DC (2007) Trehalose, a novel mTOR-independent autophagy enhancer, accelerates the clearance of mutant huntingtin and alpha-synuclein. J Biol Chem 282:5641–5652

Sava I, Davis LJ, Gray SR, Bright NA, Luzio JP (2024) Reversible assembly and disassembly of V-ATPase during the lysosome regeneration cycle. Mol Biol Cell 35:ar63

Saxton RA, Sabatini DM (2017) mTOR signaling in growth, metabolism, and disease. Cell 168:960–976

Schnupf P, Portnoy DA (2007) Listeriolysin O: a phagosome-specific lysin. Microbes Infect 9:1176–1187

Schrezenmeier E, Dörner T (2020) Mechanisms of action of hydroxychloroquine and chloroquine: implications for rheumatology. Nat Rev Rheumatol 16:155–166

Serre K, Bénézech C, Desanti G, Bobat S, Toellner K-M, Bird R, Chan S, Kastner P, Cunningham AF, MacLennan ICM et al (2011) Helios is associated with CD4 T cells differentiating to T helper 2 and follicular helper T cells in vivo independently of Foxp3 expression. PLoS ONE 6:e20731

Settembre C, Fraldi A, Medina DL, Ballabio A (2013) Signals from the lysosome: a control centre for cellular clearance and energy metabolism. Nat Rev Mol Cell Biol 14:283–296

Settembre C, Zoncu R, Medina DL, Vetrini F, Erdin S, Erdin S, Huynh T, Ferron M, Karsenty G, Vellard MC et al (2012) A lysosome-to-nucleus signalling mechanism senses and regulates the lysosome via mTOR and TFEB. EMBO J 31:1095–1108

Skowyra ML, Schlesinger PH, Naismith TV, Hanson PI (2018) Triggered recruitment of ESCRT machinery promotes endolysosomal repair. Science 360:eaar5078

So AK, Martinon F (2017) Inflammation in gout: mechanisms and therapeutic targets. Nat Rev Rheumatol 13:639–647

Stancu I-C, Cremers N, Vanrusselt H, Couturier J, Vanoosthuyse A, Kessels S, Lodder C, Brône B, Huaux F, Octave J-N et al (2019) Aggregated Tau activates NLRP3-ASC inflammasome exacerbating exogenously seeded and non-exogenously seeded Tau pathology in vivo. Acta Neuropathol 137:599–617

Steger M, Tonelli F, Ito G, Davies P, Trost M, Vetter M, Wachter S, Lorentzen E, Duddy G, Wilson S et al (2016) Phosphoproteomics reveals that Parkinson's disease kinase LRRK2 regulates a subset of Rab GTPases. eLife 5:e12813

Swanson JA, Watts C (1995) Macropinocytosis. Trends Cell Biol 5:424–428

Tan JX, Finkel T (2022) A phosphoinositide signalling pathway mediates rapid lysosomal repair. Nature 609:815–821

Tanaka Y, Guhde G, Suter A, Eskelinen EL, Hartmann D, Lüllmann-Rauch R, Janssen PM, Blanz J, von Figura K, Saftig P (2000) Accumulation of autophagic vacuoles and cardiomyopathy in LAMP-2-deficient mice. Nature 406:902–906

Terebiznik MR, Vazquez CL, Torbicki K, Banks D, Wang T, Hong W, Blanke SR, Colombo MI, Jones NL (2006) Helicobacter pylori VacA toxin promotes bacterial intracellular survival in gastric epithelial cells. Infect Immun 74:6599–6614

Terkeltaub R (2017) What makes gouty inflammation so variable? BMC Med 15:158

Thurston TLM, Wandel MP, von Muhlinen N, Foeglein Á, Randow F (2012) Galectin-8 targets damaged vesicles for autophagy to defend cells against bacterial invasion. Nature 482:414–418

van Olst L, Verhaege D, Franssen M, Kamermans A, Roucourt B, Carmans S, Ytebrouck E, van der Pol SMA, Wever D, Popovic M et al (2020) Microglial activation arises after aggregation of phosphorylated-tau in a neuron-specific P301S tauopathy mouse model. Neurobiol Aging 89:89–98

Wang A, Chen C, Mei C, Liu S, Xiang C, Fang W, Zhang F, Xu Y, Chen S, Zhang Q et al (2024a) Innate immune sensing of lysosomal dysfunction drives multiple lysosomal storage disorders. Nat Cell Biol 26:219–234

Wang Q, Wolf A, Ozkan S, Richert L, Mely Y, Chasserot-Golaz S, Ory S, Gasman S, Vitale N (2023a) V-ATPase modulates exocytosis in neuroendocrine cells through the activation of the ARNO-Arf6-PLD pathway and the synthesis of phosphatidic acid. Front Mol Biosci 10:1163545

Wang X, Espadas J, Wu Y, Cai S, Ge J, Shao L, Roux A, Camilli PD (2023b) Membrane remodeling properties of the Parkinson's disease protein LRRK2. Proc Natl Acad Sci USA 120:e2309698120

Wang X, Xu P, Bentley-DeSousa A, Hancock-Cerutti W, Cai S, Johnson BT, Tonelli F, Talaia G, Alessi DR, Ferguson SM et al (2024b) Lysosome damage triggers acute formation of ER to lysosomes membrane tethers mediated by the bridge-like lipid transport protein VPS13C. Preprint at https://www.biorxiv.org/content/biorxiv/early/2024/06/08/2024.06.08.598070.full.pdf

Westman J, Moran G, Mogavero S, Hube B, Grinstein S (2018) Candida albicans hyphal expansion causes phagosomal membrane damage and luminal alkalinization. mBio 9:e01226–18

Wilke S, Krausze J, Büssow K (2012) Crystal structure of the conserved domain of the DC lysosomal associated membrane protein: implications for the lysosomal glycocalyx. BMC Biol 10:62

Wolfmeier H, Radecke J, Schoenauer R, Koeffel R, Babiychuk VS, Drücker P, Hathaway LJ, Mitchell TJ, Zuber B, Draeger A et al (2016) Active release of pneumolysin prepores and pores by mammalian cells undergoing a Streptococcus pneumoniae attack. Biochim Biophys Acta 1860:2498–2509

Wu Y, Shao W, Todd TW, Tong J, Yue M, Koga S, Castanedes-Casey M, Librero AL, Lee CW, Mackenzie IR et al (2021) Microglial lysosome dysfunction contributes to white matter pathology and TDP-43 proteinopathy in GRN-associated FTD. Cell Rep. 36:109581

Xu H, Ren D (2015) Lysosomal physiology. Annu Rev Physiol 77:57–80

Yagnik DR, Hillyer P, Marshall D, Smythe CD, Krausz T, Haskard DO, Landis RC (2000) Noninflammatory phagocytosis of monosodium urate monohydrate crystals by mouse macrophages. Implications for the control of joint inflammation in gout. Arthritis Rheum 43:1779–1789

Yang EH, Rode J, Howlader MA, Eckermann M, Santos JT, Armada DH, Zheng R, Zou C, Cairo CW (2017) Galectin-3 alters the lateral mobility and clustering of β1-integrin receptors. PLoS ONE 12:e0184378

Ye T, Wang C, Yan J, Qin Z, Qin W, Ma Y, Wan Q, Lu W, Zhang M, Tay FR et al (2023) Lysosomal destabilization: a missing link between pathological calcification and osteoarthritis. Bioact Mater 34:37–50

Zhang Q, Li Y, Jian Y, Li M, Wang X (2023) Lysosomal chloride transporter CLH-6 protects lysosome membrane integrity via cathepsin activation. J Cell Biol 222:e202210063

## Acknowledgements

SAF is the recipient of a Canada Research Chair and was supported by grants PJT-169180 and PJT-190244 from the Canadian Institutes of Health Research (CIHR). OS is supported by a fellowship award from the Canadian Allergy Asthma and Immunology Foundation (CAAIF) and Immunodeficiency Canada, as well as by the Hospital for Sick Children Transition Clinician Scientist Program (Department of Paediatrics). Figures were generated with a paid for licence of Biorender.

## Author contributions

**Ori Scott**: Conceptualization; Writing—original draft; Writing—review and editing. **Ekambir Saran**: Writing—original draft. **Spencer A Freeman**: Conceptualization; Writing—original draft; Writing—review and editing.

## Disclosure and competing interests statement

The authors declare no competing interests.

