## [Peer Review File · EMBO Reports]

The spectrum of lysosomal stress and damage responses: from mechanosensing to inflammation

Spencer Freeman, Ori Scott, and Ekambir Saran

Corresponding author(s): Spencer Freeman (spencer.freeman@sickkids.ca)

Review Timeline:

Submission Date:	7th Oct 24
Editorial Decision:	18th Nov 24
Revision Received:	26th Nov 24
Editorial Decision:	31st Jan 25
Revision Received:	7th Feb 25
Accepted:	12th Feb 25

Editor: *Martina Rembold*

Transaction Report:

Dear Spencer,

Thank you once more for the submission of your review to EMBO Reports. I have now received two referee reports that are copied below. As you will see, both referees state that your manuscript is interesting and timely, but they have also several suggestions to improve it, which I kindly ask you to address.

I would further suggest expanding the discussion on pathogens, phagocytes and lysosomal stress/function, if possible, as I feel that this could contribute a very distinct angle that will set your review apart from earlier reviews on lysosomal stress and repair (see also the recent review from Tan & Finkel in EMBO Reports with a focus on aging and senescence, <https://doi.org/10.15252/embr.202357265>).

I further felt that you could add an additional figure for the section on the beneficial effects of lysosomal damage. We can accommodate up to 5 figures.

Please also address the following editorial points:

- Please add a 'Disclosure and competing interests statement'. For more information see <https://www.embopress.org/page/journal/14693178/authorguide#conflictsofinterest>

- Please add up to five keywords to the title page.

- Please provide a section called "Box1: in need of answers". This section is a bullet point list of open questions in the field.

When submitting your revised manuscript, we will require a Microsoft Word file (.docx) of the revised manuscript text including detailed figure legends (at the very end), but without the figures.

- Please provide the final figures as separate, high resolution files as .pdf, .eps, .tif, or .jpg (one file per figure). Please finalize the drafts provided and make sure they accurately illustrate the key scientific concepts that you wish to show.

Please also note the following general advice on review figures:

- If there are certain aspects of your figure draft that are based upon assumptions or where the scientific data remains ambiguous (for example, schematically depicting a presumed direct protein-protein interaction, protein shape or subcellular localizations etc.) please add a comment so that we can work with you on an accurate depiction. Please ensure the directionality and nature of interactions is presented accurately.

- If the figure or single panels of the figure have been adapted from a published figure, please add this information to the figure legend (e.g., 'Adapted from...' or 'Based on...'). The editor will discuss if a reference and permission will be necessary

- Please only re-use figures or parts of a figure if this is essential for understanding the concept communicated. Often a reference to a previous paper will suffice. If the figure contains re-used images or elements of images, including schematics, micrographs or photos, please make sure that you have the permission/license to publish it (this also applies to your own previous work, if the journal you published in retains copyright. Certain 'creative commons' open access licenses, such as CC-BY 4.0, allow re-use without additional formal permissions). All re-used material must be explicitly cited.

- If you use an image data base for scientific iconography (e.g., BioRender), please let us know if you have a license that allows for publication in an academic journal. Often authors use misleading iconography for expedience. Please ensure the information shown is scientifically accurate. If in doubt, please discuss with the editor.

Thank you again for writing this article for us. I look forward to seeing a revised version of your manuscript when it is ready. As for timing, would it be possible for you to submit the revised version by end of January? But let me know if you need more time.

With best wishes,

Martina

=====

Referee #1:

This is a well written and extremely timely review, highlighting both factors that contribute to lysosomal stress and cellular responses to lysosomal damage. Different from most recent reviews on this subject, there is emphasis on the importance of lysosomal stress in human physiology and pathology. This will be a useful and well received reference for the field.

I have little to suggest on what is already a comprehensive review of the topic.

p. 12 - line 281 missing full reference

p. 13 line 308 incomplete sentence

p. 15 lines 356-359. Is there a reference indicating that galectin-3 forms a lattice to create a diffusional barrier?

p. 16 discussion re: galectins - is there a reason to think that galectins 3 & 8 are unique or just the two that have been best studied in this context? Perhaps worth noting that there are others?

p. 16 lines 380-381 - Since ubiquitination itself does not need to be slow, what is rate limiting to make initiation of lysophagy slow?

Referee #2:

This is a thorough review on the topic of lysosome stress and damage responses. Given that this topic is capturing increasing attention, it is also timely. The structure is logical and important topics are well covered. I think that it could be improved by relying less on citing review articles and more citations of the primary literature that supports key points. Below I have provided additional specific suggestions for strengthening the review.

1. The concept of the "glycocalyx" is important for this review. It would be helpful to have citations of the underlying primary literature that established the existence and function of the glycocalyx. The Wilkes et al reference on line 47 does not meet this expectation.
2. There is an over reliance on review articles to support many ideas that are presented. It would be more helpful to cite the primary literature. For example, 2 of the 3 references in line 101 are to reviews. Also, line 109 regarding crystals in gout cites a review. Also, lines 314 and 315 that refer to the role for LRRK2 in promoting lysosome recruitment of AP-3 and actin nucleators support these claims with a review article. This is followed on line 316 by citation of another review article. Lines 481-484 regarding alum and antigen presentation is supported by a review article. 486-489 also deals with alum and reference to a review article. These are examples of a larger pattern that sometimes makes it hard to find and evaluate the data that underlies interesting ideas that are presented.
3. The authors should seek to support key points with references. For example, lines 378-379 propose (without references) that the degree of damage required to initiate lysophagy is substantial but does not define what exactly constitutes substantial damage. Lines 390-392 link (without references) persistent lysosome damage to diseases.
4. Lines 43-44 mention co-factors that extract lipids from membranes within lysosomes but do not provide a reference.
5. Lines 220-231 suggest that ESCRTs deform lysosomal membranes to ensure a reservoir of membrane that can help to accommodate increases in lysosome size. However, there is normally not a significant amount of ESCRT proteins at lysosomes until after damage is induced. To serve as a buffer, the reservoir would need to be present prior to any damage. These reservoirs are also implied in Figure 1 and stated in Figure 2. However, there is a lack of supporting citations or explanation of the underlying data that directly demonstrate that this is the case.
6. Line 301: Similar to the above concern, it is not clear how recruitment of ESCRTs in response to Ca²⁺ release would help to generate membrane reserves that help with response to this acute stress. They do not generate any new membrane and there are no references provided that demonstrate that they provide a stable reserve as opposed to transient buds that progress to intraluminal vesicles.
7. Line 281, Lenk reference details missing
8. Line 288, NH₄⁺ repeated.
9. Line 308-311: These papers that propose regulated interactions between the v-ATPase and GTPases are relatively old. Have these proposed interactions been replicated in subsequent studies?
10. Line 340: it would be helpful to elaborate on evidence that ESCRT-dependent intraluminal vesicle budding selectively sorts factors that damage lysosomes.
11. Lines 353-354: A reference to support the idea presented here would be helpful.

12. Lines 409-411: Sardiello et al, 2009 were the first to show that solute (sucrose) accumulation in lysosomes leads to TFEB accumulation in the nucleus.

13. Is there a reference that provides evidence that ceramides help to promote ESCRT-dependent inward budding?

14. Lines 595-601: How would lysosome permeabilization lead to cathepsin secretion rather than their release into the cytoplasm?

For clarity, we have copied **editorial** and **reviewer comments** in blue and respond to each in black. In the manuscript, we have indicated sections that have been revised in red, we highlight added references in cyan and we indicate the line numbers in which these changes can be found in the text below.

Editor comments:

I would further suggest expanding the discussion on pathogens, phagocytes and lysosomal stress/function, if possible, as I feel that this could contribute a very distinct angle that will set your review apart from earlier reviews on lysosomal stress and repair (see also the recent review from Tan & Finkel in EMBO Reports with a focus on aging and senescence, <https://doi.org/10.15252/embr.202357265>).

We appreciate the importance of ensuring this review is recognized by the field as being different from that of Tan and Finkel; its contents already contained many novel aspects and viewpoints including the notions of protective mechanisms of lysosomes (i.e. the glycocalyx), the beneficial effects of lysosome damage in the context of immune responses, and the stress placed on the endocytic pathway upon host-pathogen encounters, which was recognized by reviewers. However, to further highlight and expand on these parts of the review, we have now done the following: *i*) we now include an expansive sub-section on the role of pathogens in lysosomal pathology, which elaborates on pore-forming toxins and contains added information on virulence factors produced by Gram positive bacteria (located on pages 6-7), *ii*) we have included additional examples of bacterial species and their secreted pore-forming toxins in Figure 4 (see below), *iii*) we have added “host-pathogen” and “pore-forming toxins” to the key words, and *iv*) we have expanded on other novel aspects of the review, including the protective glycocalyx of lysosomes, which we describe in detail in the response to reviewers below.

I further felt that you could add an additional figure for the section on the beneficial effects of lysosomal damage. We can accommodate up to 5 figures.

This is a wonderful suggestion. Very recently, two papers were published in *Science* and *Science Immunology* on the topic of endogenous damaging protein complexes, perforin-2 and APOL7C. We cite this work and have now added an entirely new panel to Figure 4 below to include details of the mechanisms involved. This panel is entitled “Endogenous means of membrane perforation”.

New Figure 4:

Please also address the following editorial points:

- Please add a 'Disclosure and competing interests statement'. For more information see

<https://www.embopress.org/page/journal/14693178/authorguide#conflictofinterest>

A disclosure and competing interests statements has been added to the title page.

- Please add up to five keywords to the title page.

Five key words have been added to the title page.

- Please provide a section called "Box1: in need of answers". This section is a bullet point list of open questions in the field.

This section has now been added.

When submitting your revised manuscript, we will require a Microsoft Word file (.docx) of the revised manuscript text including detailed figure legends (at the very end), but without the figures.

Done

- Please provide the final figures as separate, high resolution files as .pdf, .eps, .tif, or .jpg (one file per figure). Please finalize the drafts provided and make sure they accurately illustrate the key scientific concepts that you wish to show.

Done

Please also note the following general advice on review figures:

- If there are certain aspects of your figure draft that are based upon assumptions or where the scientific data remains ambiguous (for example, schematically depicting a presumed direct protein-protein interaction, protein shape or subcellular localizations etc.) please add a comment so that we can work with you on an accurate depiction. Please ensure the directionality and nature of interactions is presented accurately.

NA

- If the figure or single panels of the figure have been adapted from a published figure, please add this information to the figure legend (e.g., 'Adapted from...' or 'Based on...'). The editor will discuss if a reference and permission will be necessary

NA

- Please only re-use figures or parts of a figure if this is essential for understanding the concept communicated. Often a reference to a previous paper will suffice. If the figure contains re-used images or elements of images, including schematics, micrographs or photos, please make sure that you have the permission/license to publish it (this also applies to your own previous work, if the journal you published in retains copyright. Certain 'creative commons' open access licenses, such as CC- BY 4.0, allow re-use without additional formal permissions). All re-used material must be explicitly cited.

NA

- If you use an image data base for scientific iconography (e.g., BioRender), please let us know if you have a license that allows for publication in an academic journal. Often authors use misleading iconography for expedience. Please ensure the information shown is scientifically accurate. If in doubt, please discuss with the editor.

We have a license for the software used (Biorender) and acknowledge this in the manuscript.

Referee #1:

This is a well written and extremely timely review, highlighting both factors that contribute to lysosomal stress and cellular responses to lysosomal damage. Different from most recent reviews on this subject, there is emphasis on the importance of lysosomal stress in human physiology and pathology. This will be a useful and well received reference for the field.

We thank the reviewer for their positive view of the manuscript and for taking the time to review it. Below, we have attended to their comments.

I have little to suggest on what is already a comprehensive review of the topic.

p. 12 - line 281 missing full reference

The full reference is now added (see line 318).

p. 13 line 308 incomplete sentence

This sentence is now correct (see line 345-348).

p. 15 lines 356-359. Is there a reference indicating that galectin-3 forms a lattice to create a diffusional barrier?

In lines 395-400 we now clarify that the notion of galectin-3 as forming a lattice-like diffusional barrier has been extrapolated from what is known regarding its interaction with glycosylated proteins when deposited on the plasma membrane. Remarkably, many of the physicochemical aspects of the glycocalyx have yet to be explored for lysosomes, so we have highlighted this knowledge gap while also emphasizing its clear importance.

p. 16 discussion re: galectins - is there a reason to think that galectins 3 & 8 are unique or just the two that have been best studied in this context? Perhaps worth noting that there are others?

In line 419-421, we have now added clarification that while galectins-3 and -8 are the best studied in the context of the lysosomal damage response, other galectins may well be involved. Galectin-3 shows prominent expression in phagocytic cells, which we mention, suggesting it may be particularly important in responses of macrophages.

p. 16 lines 380-381 - Since ubiquitination itself does not need to be slow, what is rate limiting to make initiation of lysophagy slow?

In lines 422-428, we clarify that lysophagy is slow not just due to the ubiquitination step per se, but also the need for recognition of ubiquitination and phagophore formation. This is entirely relative to other mechanisms that are near instantaneous (i.e. ion transport). We state:

“Importantly, the degree of luminal exposure required to activate lysophagy is substantial, as gaps of even up to 200 nm appear to be amenable to repair (Ellison et al., 2020). This allows for differential activation of early repair pathways following smaller rupture. Additionally, the need for ubiquitination as a regulatory gatekeeper, its recognition by receptors the ensuing phagophore formation makes lysophagy relatively slow. This allows rapid repair pathways to tip the scale in favour of salvage in cases of modest damage (Fujita et al., 2013; Maejima et al., 2013).”

Referee #2:

This is a thorough review on the topic of lysosome stress and damage responses. Given that this topic is capturing increasing attention, it is also timely. The structure is logical and important topics are well covered. I think that it could be improved by relying less on citing review articles and more citations of the primary literature that supports key points. Below I have provided additional specific suggestions for strengthening the review.

We thank the reviewer for their comprehensive comments below, their general attention to detail, as well as their positive impression of the manuscript. To address their comments, we have made extensive additions to the reference list to include more citations to the primary literature and we further qualify select segments of the review, all described below.

1. The concept of the "glycocalyx" is important for this review. It would be helpful to have citations of the underlying primary literature that established the existence and function of the glycocalyx. The Wilkes et al reference on line 47 does not meet this expectation.

We have added further references in lines 53-54 which include original descriptions of the glycosylation and protective features of Lamp family proteins and the lysosomal glycocalyx:

Carlsson, S. R., Roth, J., Piller, F., & Fukuda, M. (1988). Isolation and characterization of human lysosomal membrane glycoproteins, h-lamp-1 and h-lamp-2. Major sialoglycoproteins carrying polylectosaminoglycan. *Journal of Biological Chemistry*

Kundra, R., & Kornfeld, S. (1999). Asparagine-linked oligosaccharides protect Lamp-1 and Lamp-2 from intracellular proteolysis. *The Journal of Biological Chemistry*

Neiss, W. F. (1984). A coat of glycoconjugates on the inner surface of the lysosomal membrane in the rat kidney. *Histochemistry*

We have also added glycocalyx to the key words.

2. There is an over reliance on review articles to support many ideas that are presented. It would be more helpful to cite the primary literature.

In many instances, we have now increased citations to the primary literature where appropriate. This includes those cases that the reviewer has brought to our attention below.

For example, 2 of the 3 references in line 101 are to reviews.

We now use the following references to support particulate uptake as invoking an inflammatory response in macrophages:

Becker, L., Gharib, S. A., Irwin, A. D., Wijsman, E., Vaisar, T., Oram, J. F., & Heinecke, J. W. (2010). A macrophage sterol-responsive network linked to atherogenesis. *Cell Metabolism*

Dostert, C., Pétrilli, V., Van Bruggen, R., Steele, C., Mossman, B. T., & Tschopp, J. (2008). Innate immune activation through Nalp3 inflammasome sensing of asbestos and silica. *Science*

Duewell, P., Kono, H., Rayner, K. J., Sirois, C. M., Vladimer, G., Bauernfeind, F. G., Abela, G. S., Franchi, L., Nuñez, G., Schnurr, M., Espevik, T., Lien, E., Fitzgerald, K. A., Rock, K. L., Moore, K. J., Wright, S. D., Hornung, V., & Latz, E. (2010). NLRP3 inflammasomes are required for atherogenesis and activated by cholesterol crystals. *Nature*

Martin, W. J., Walton, M., & Harper, J. (2009). Resident macrophages initiating and driving inflammation in a monosodium urate monohydrate crystal-induced murine peritoneal model of acute gout. *Arthritis and Rheumatism*

Rajamäki, K., Lappalainen, J., Öörni, K., Välimäki, E., Matikainen, S., Kovanen, P. T., & Eklund, K. K. (2010). Cholesterol Crystals Activate the NLRP3 Inflammasome in Human Macrophages: A Novel Link between Cholesterol Metabolism and Inflammation. *PLOS ONE*

Also, line 109 regarding crystals in gout cites a review.

We have now added the following reference:

Martin, W. J., Walton, M., & Harper, J. (2009). Resident macrophages initiating and driving inflammation in a monosodium urate monohydrate crystal-induced murine peritoneal model of acute gout. *Arthritis and Rheumatism*

Also, lines 314 and 315 that refer to the role for LRRK2 in promoting lysosome recruitment of AP-3 and actin nucleators support these claims with a review article. This is followed on line 316 by citation of another review article.

We now state that:

“LRRK2 can then phosphorylate and activate Rabs (Dou et al., 2024; Steger et al., 2016), and also recruit ESCRT proteins (Herbst et al., 2020), adaptor proteins (AP-3) (Kuwahara et al., 2016), and actin nucleators to drive membrane remodeling (Bonet-Ponce et al., 2020; Civiero et al., 2018; Kim et al., 2018; X. Wang et al., 2023).”

Lines 481-484 regarding alum and antigen presentation is supported by a review article. 486-489 also deals with alum and reference to a review article.

We now state that:

“Alum is a common vaccine adjuvant which stimulates inflammatory dendritic cells (Kool et al., 2008), promotes antigen-induced CD4 T cell differentiation and proliferation (Grun & Maurer, 1989; Mannhalter et al., 1985; Serre et al., 2011), and generation of predominantly IgG1 antibodies (Lindblad et al., 1997).”

These are examples of a larger pattern that sometimes makes it hard to find and evaluate the data that underlies interesting ideas that are presented.

We appreciate the comment and have integrated many other references to the primary literature across the manuscript. We have kept references to seminal narrative reviews, as well as reviews which proposed new insights or ideas based on synthesis of primary literature.

3. The authors should seek to support key points with references. For example, lines 378-379 propose (without references) that the degree of damage required to initiate lysophagy is substantial but does not define what exactly constitutes substantial damage. Lines 390-392 link (without references) persistent lysosome damage to diseases.

We now qualified the degree of damage amenable to repair as proposed in literature. References have been added linking lysosomal stress or damage to disease.

We state that:

“Importantly, the degree of luminal exposure required to activate lysophagy is substantial, as gaps of even up to 200 nm appear to be amenable to repair (Ellison et al., 2020). This allows for differential activation of early repair pathways following smaller rupture. Additionally, the need for ubiquitination as a regulatory gatekeeper, its recognition by receptors the ensuing phagophore formation makes lysophagy relatively slow. This allows rapid repair pathways to tip the scale in favour of salvage in cases of modest damage (Fujita et al., 2013; Maejima et al., 2013).”

4. Lines 43-44 mention co-factors that extract lipids from membranes within lysosomes but do not provide a reference.

We now state that (lines 51-52):

“Lysosomes also harbour lipid-binding proteins that serve as co-factors for luminal hydrolases by extracting lipids and solubilizing membranous contents (Fürst & Sandhoff, 1992; Matsuda et al., 2004; Morimoto et al., 1989).”

5. Lines 220-231 suggest that ESCRTs deform lysosomal membranes to ensure a reservoir of membrane that can help to accommodate increases in lysosome size. However, there is normally not a significant amount of ESCRT proteins at lysosomes until after damage is induced. To serve as a buffer, the reservoir would need to be present prior to any damage. These reservoirs are also implied in Figure 1 and stated in Figure 2. However, there is a lack of supporting citations or explanation of the underlying data that directly demonstrate that this is the case.

We thank the reviewer for this comment. In principle, the ongoing activity of ESCRTs to late endosomes (for example, to produce inward vesicles) could provide membrane reserves provided increases in tension occur before scission of the vesicles. That ESCRTs can protect lysosomes from overt damage is supported by a recent paper by the Hanson lab (2024), which is referenced in lines 259-261. In this study, the Ca²⁺ recruitment of ESCRT can in principle be fulfilled by cation channels resident to the

compartment which respond to increased tension without there being damage to the limiting membrane. While it may not be clear how ESCRT is protective in such instances, it is possible that ESCRT or any curvature generating complex could help to facilitate membrane fusion or organelle contact sites under high tension/stress when the bulging intermediate formation requires local deformation beyond what it provided by SNAREs.

6. Line 301: Similar to the above concern, it is not clear how recruitment of ESCRTs in response to Ca²⁺ release would help to generate membrane reserves that help with response to this acute stress. They do not generate any new membrane and there are no references provided that demonstrate that they provide a stable reserve as opposed to transient buds that progress to intraluminal vesicles.

We are equally puzzled by the protective role of ESCRT complexes but believe that this finding is important to highlight and is likely quite important. We now state on line 259: “Importantly, recent work demonstrated that the Ca²⁺-dependent recruitment of ESCRT to lysosomes experiencing increased tension prevented their rupture (W. Chen et al., 2024). While the mechanism(s) by which ESCRT complexes protect the membrane from damage are not yet clear, it is conceivable that deformation of the membrane under high stress/tension facilitates contacts with other organelles and/or fusion.”

7. Line 281, Lenk reference details missing

The full reference details have now been inserted (see line 318)

8. Line 288, NH₄⁺ repeated.

This is corrected which can now be located on line 325

9. Line 308-311: These papers that propose regulated interactions between the v-ATPase and GTPases are relatively old. Have these proposed interactions been replicated in subsequent studies?

Thank you for pointing this out. There are some recent examples in which the V-ATPase can directly recruit GEFs. We now state on line 345:

“For example, lysosomal alkalization, sensed directly by the V-ATPase, leads to conformational changes to the pump followed by the recruitment of GTPases in the Rab and Arf families from the cytosol along with their respective guanine-exchange factors (GEFs) (Hurtado-Lorenzo et al., 2006; Maranda et al., 2001; Matsumoto et al., 2022; Q. Wang et al., 2023).”

10. Line 340: it would be helpful to elaborate on evidence that ESCRT-dependent intraluminal vesicle budding selectively sorts factors that damage lysosomes.

We again thank the reviewer for this very good point. While the selective removal of damaging factors by ESCRT complexes has been established for the plasma membrane, this has not been shown for lysosomes. In lines 380-383, we clarify this: “In the plasma membrane, ECSRT has been shown to selectively remove damaging agents such as pore-forming toxins, along with wounded membrane (Jimenez et al., 2014; Wolfmeier et al., 2016). However, this phenomenon remains to be demonstrated specifically in endolysosomes.”

11. Lines 353-354: A reference to support the idea presented here would be helpful.

A reference has been added in line 394.

12. Lines 409-411: Sardiello et al, 2009 were the first to show that solute (sucrose) accumulation in lysosomes leads to TFEB accumulation in the nucleus.

This reference is now added to line 457.

13. Is there a reference that provides evidence that ceramides help to promote ESCRT-dependent inward budding?

This is the idea put forward by the Holthius group and has also been shown for the plasma membrane by others. We now include reference to Niekamp et al. 2022 on line 394.

14. Lines 595-601: How would lysosome permeabilization lead to cathepsin secretion rather than their release into the cytoplasm?

While there are reports of circulating cathepsins upon lysosomal cell death, this is likely owed to secondary lytic cell death. We have removed discussion of this as it is undoubtedly secondary to the damage of the lysosomes.

Dear Spencer,

Please accept my sincere apologies regarding the delay in proceeding with the publication of your review article.

I have meanwhile read your response to the referee concerns and I have read the review article again. I concur with the referees that the article is a beautiful read and a very informative and comprehensive overview on lysosomal damage and stress response. I did not consider further developmental editing necessary, but have a few editorial requests and a few suggestions, mostly regarding the figures.

Please address these minor concerns (listed below my signature) and once you have resubmitted the revised review article, we will swiftly proceed with its publication. It is really a nice and very timely review that will be appreciated by the field.

Kind regards,

Martina

Martina Rembold, PhD
Senior Editor
EMBO Reports

=====

Minor editorial requests and suggestions:

- Please remove the figures from the manuscript text file. The individual figure files are sufficient.
- Please place the Conflict of Interest Statement between the Acknowledgements and the References.
- We have a word limit of 175 for the Abstract. I think we can be a bit flexible here with the review, but if you could shorten it a little bit, that would be great.
- Funding information in the Acknowledgements in the online manuscript tracking system must be congruent. We are currently missing in the online system: PJT-190244; Canadian Allergy Asthma and Immunology Foundation (CAAIF) and Immunodeficiency Canada; the Hospital for Sick Children Transition Clinician Scientist Program (Department of Paediatrics)
- References: et al needs to be used after 10 author names; DOIs should only be used for preprints and datasets that have not been published yet.
- Biorxiv citations: please add (preprint: Name et al, Year) in the text and the label [PREPRINT] in the reference list, after the DOI.
- I wonder whether you really need to show the lysosome schematics from Figure 1 (Protection, damage sensing and repair) again in Figures 2 and 4. It seems repetitive and not adding much to the figure. Maybe you could add subtitles to the legend of Figure 2 that repeat the keywords "Curvature/membrane reserves" and "glycolax". Similar thoughts apply to Figure 4. Removing the schematic image on the left, would give the three panels on the right more space.
- Figure 3 shows "PI4K2A" and "ORPs", which have not been described in the main text. You discuss phosphatidylinositol phosphates and PIKfyve but not the role of PI4K2A in this context. The arrows in the figure imply that it is required to recruit VPS13C, ATG2, and ORPs. It might be good to add a sentence or two in the text (or the legend). In addition, the meaning of ORPs has not been defined, unless I missed it?
- Figure 4: In the text you first describe exogenous sources of damage, followed by repair mechanisms (ESCRT). Thus, the top and bottom panels of Figure 4. The middle panel "Endogenous means of membrane perforation" is only described in the section on "Beneficial effects of controlled lysosomal damage". What about moving this panel to a dedicated figure on "Beneficial effects of controlled, endogenous, lysosomal damage", or alike? We still have room for figures. Please also add a short statement in the legend to explain the action of NOX2 for those readers who are not familiar with it.
- The paragraph "In need of answers" is called "Box1. In need of answers". Please adjust the heading and call the text on in vivo imaging of lysosomal damage "Box 2".
- Line 417 to 428, uses a lot of adverbs in consecutive sentences. "Interestingly", "Notably", "Importantly", and "Additionally". I suggest removing or at least reducing these.

All editorial and formatting issues were resolved by the authors.

Dr. Spencer Freeman
The Hospital for Sick Children
686 Bay St
Toronto, ON M5G0A4
Canada

Dear Spencer,

Thank you very much for submitting your revised review. I am very pleased to inform you that your manuscript has been accepted for publication in EMBO reports. It was a pleasure to work with you on this review and my congratulations to a review article that will be of great interest for the field.

Your manuscript will be processed for publication by EMBO Press. It will be copy edited and you will receive page proofs prior to publication.

When you receive an email from Springer Nature asking you to sign your license agreement, please enter the following code on the payment screen which should remove any charges due: NDI3MDKZMW.

Should you experience any difficulty, please email publishing@embo.org.

If you have any questions, please do not hesitate to contact me. Thank you for your contribution to EMBO Reports.

Kind regards,

Martina
